# The representation of context in mouse hippocampus is preserved despite neural drift

Alexandra T. Keinath 1✉, Coralie-Anne Mosser 1 & Mark P. Brandon 1✉

The hippocampus is thought to mediate episodic memory through the instantiation and reinstatement of context-specific cognitive maps. However, recent longitudinal experiments have challenged this view, reporting that most hippocampal cells change their tuning properties over days even in the same environment. Often referred to as *neural* or *representational drift*, these dynamics raise questions about the capacity and content of the hippocampal code. One such question is whether and how these long-term dynamics impact the hippocampal code for context. To address this, we image large CA1 populations over more than a month of daily experience as freely behaving mice participate in an extended geometric morph paradigm. We find that long-timescale changes in population activity occur orthogonally to the representation of context in network space, allowing for consistent readout of contextual information across weeks. This population-level structure is supported by heterogeneous patterns of activity at the level of individual cells, where we observe evidence of a positive relationship between interpretable contextual coding and long-term stability. Together, these results demonstrate that long-timescale changes to the CA1 spatial code preserve the relative structure of contextual representation.

[1] Department of Psychiatry, Douglas Hospital Research Centre, McGill University, 6875 Boulevard LaSalle, Verdun, QC H4H 1R3, Canada.
✉email: atkeinath@gmail.com; mark.brandon@mcgill.ca

Hippocampal subregion CA1 represents a mixture of external and internal cues—including the shape of the environment[1], visual landmarks[2,3], objects[4], task-relevant information[5], and past experience[6–9]—through changes to the spatial tuning properties of its principal cells[10]. Often collectively referred to as a *cognitive* or *hippocampal map*[11], this code is hypothesized to support spatial and episodic memory by reinstating content at later times, for example by reinstantiating the same map across repeated visits to the same environment. However, recent longitudinal experiments in mice have challenged this view[12–16]. These studies report that the majority of cells change their spatial tuning properties, such as firing rates[13] and field locations[14], over a timescale of days, yielding maps of the same environment that, across time, are as different as maps between environments when assessed by various population-level measures. Despite these population-level dynamics, however, a minority of cells can maintain their spatial tuning properties across a months-long timescale under at least some circumstances.

These long-term dynamics raise a variety of questions about the representational capacity and content of the hippocampal code[17], questions which might be resolved in multiple ways. On one hand, some suggest that these dynamics reflect the unavoidable biological realities of this code rather than its representational content[17–19]. That is, while the hippocampal code might serve a traditional mapping function, it is constrained by inherently variable and plastic circuits. Thus, this account claims, its representation will drift over time, i.e. the representation will change even in the absence of changes to the content of the representation. On the other hand, it is possible that these dynamics instead faithfully reflect changes in the content of the representation. Although the environment remains the same across repeated visits in a spatial sense, each visit differs due to a variety of factors such as time, intervening experience, and idiosyncratic external cues which might elude the control of the experimenter. Given the responses of the CA1 spatial code to such diverse content on short timescales, it is thus plausible that the long-term dynamics are not a biological accident but instead accurately recapitulate the heterogenous dynamics of its content. Yet another possibility is that these dynamics point toward a necessary revision of function beyond a mapping framework, for example one that emphasizes the temporal structure over the spatial correlates of this code.

Central to the debate between these views is the relationship between the representational capacity and the long-term dynamics of the CA1 code. While prior work has informatively characterized long-term dynamics within the same environment and across two highly distinct environments[13–16], such work is in some ways limited in its ability to speak to this relationship. Repeated recordings in the same environment characterize only effects occurring within that spatial context, whereas recording in two distinct environments has yielded representations which begin and remain orthogonal throughout, and therefore any changes to the relative representational structure cannot be characterized beyond orthogonality.

To address this knowledge gap, we imaged large CA1 populations over more than a month of daily experience as freely behaving mice explored six differently-shaped environments in an extended geometric morph paradigm[8]. This paradigm elicited partially-correlated population-level maps, allowing us to characterize how the relative representational structure evolved over long timescales. We found that individual cells represented spatial context through heterogenous but stereotyped changes to their spatial tuning properties which led to a dynamic attractor-like population response. Characterizing the full representational structure of all sessions revealed that long-timescale changes in population activity occurred orthogonally to the representation of context in network space. Thus, despite continued representational changes over the course of the experiment, the relative representational structure of the environments was preserved, allowing for consistent readout of contextual information across weeks. These population-level dynamics were supported by heterogeneous responses at the level of single cells, with many cells exhibiting interpretable patterns of contextual coding with little evidence of drift and vice versa. Together these results demonstrate that neural drift, i.e., the long-timescale changes to the CA1 spatial code observed even across repeated visits to the same environment, does not distort the relative structure of contextual representation in mouse hippocampus.

## Results

### Assaying contextual representation across extended experience.

We recorded daily from large CA1 populations via calcium imaging (Fig. 1a) as mice freely explored open environments for 32 days in an extended version of a geometric morph paradigm (5 mice, 160 sessions total; Fig. 1b and Fig. S1). In this paradigm, mice were initially familiarized with two geometrically-distinct environments and later tested in these environments as well as four deformed (morphed) versions of these environments spanning the shapespace between the two familiar environments. On each test day, activity was recorded in a single environment, after which the mice received additional unrecorded top-up experience in the remaining familiar environment(s). The test phase began with recording both familiar environments across 2 days (order randomized across mice). Next, activity was recorded in each morph environment (in random order) over 4 days. Then the familiar environments were recorded again to bookend that morph sequence. This pattern was repeated for 32 test days at which point five full bookended morph sequences were recorded. The order of unrecorded experiences was randomized on each day. All sessions (including all unrecorded top-up experiences) lasted 20 min per environment. Only one environment was recorded per day to reduce the risk of photobleaching.

We first characterized these data within each morph sequence. Following motion correction[20], cells were segmented and calcium traces were extracted via constrained nonnegative matrix factorization[21,22] (CNMFE; Fig. S1). The rising phase of transients was extracted from the filtered calcium traces, and this binary vector was treated as the *firing rate* in all further analyses. Cell identity was tracked across recording sessions on the basis of mean imaging frame landmarks and the spatial footprints extracted by CNMFE, i.e., the spatial component of the matrix factorization which corresponds to the location and shape of each cell extracted in the field of view ($n = 4070$ cells)[23]. Although all mice continued to sample the environments thoroughly throughout the experiment (Fig. S2), to ensure that spatial sampling differences between environments and across time did not impact our findings we subsampled our data to match the spatial sampling distributions between all comparisons (see Methods)[24]. Measures of place code quality remained high across mice and throughout all sessions (Fig. S3).

We observed heterogeneous but stereotyped responses at the level of individual cells (Fig. 1c), similar to previous reports from acute versions of the geometric morph paradigm[8,9]. Some cells exhibited attractor-like properties in their spatial tuning, with either an abrupt transition in their preferred firing locations over the course of the morph sequence or the expression of a punctate field in only a contiguous subset of the shapespace. Other cells continued to fire at geometrically-similar locations across all environments spanning the shapespace. At a population level, these single-cell responses drove sigmoidal changes in mean rate

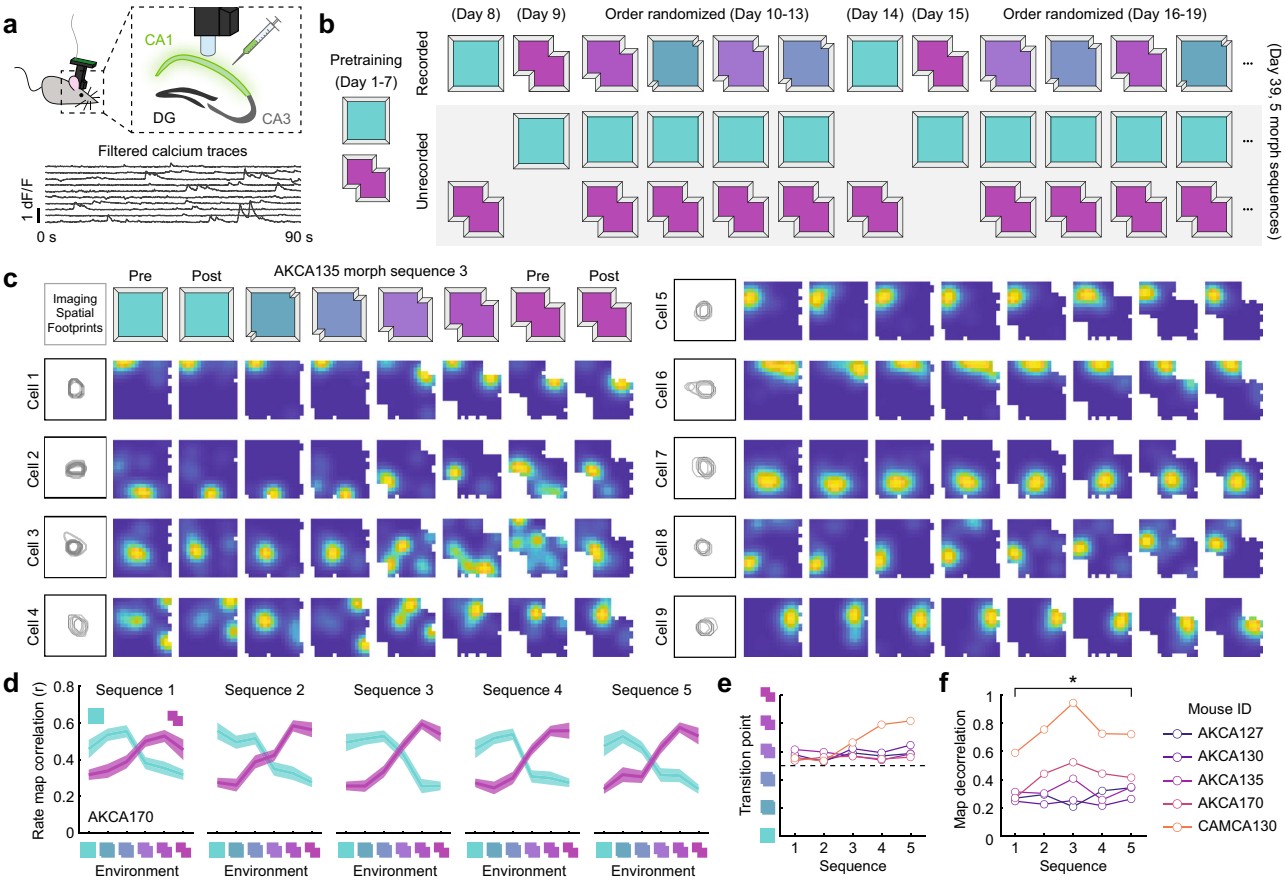

**Fig. 1 An adapted geometric morph paradigm can characterize the CA1 spatial code across extended experience. a** Schematic of the miniscope recording procedure (top) and resulting calcium traces (bottom). **b** Schematic of the behavioral paradigm. **c** Example of nine simultaneously recorded cells from mouse AKCA135 tracked across one morph sequence exhibiting a diversity of dynamics. Imaging footprints for all eight sessions are shown left of rate maps. Rate maps normalized from zero (blue) to the peak (yellow) within each session. **d** Rate map correlations between each environment and the two familiar environments for all five eight-day bookended morph sequences for one example mouse. Mouse ID in the lower left. Lines and shading denote mean ± 1 SEM across all cells whose within-session split-half reliability exceeded the 95th percentile of a shuffled distribution for at least one of the compared sessions. **e** Transition point as a function of the sequence number. **f** Decorrelation between the familiar environment rate maps as a function of the sequence number for all five mice. Familiar environment rate maps were more decorrelated in the final sequence than the initial sequence (two-tailed paired $t$-test: $t(4) = 3.12$, $p = 0.0354$). Source data provided as Source Data file. *$p < 0.05$.

map similarity across the shapespace, such that maps remained similar to the nearby familiar environment map, with an abrupt transition in similarity near the midpoint of the shapespace, echoing previous acute reports[8,9] (Fig. 1d and Fig. S4). In four of the five mice, the point of transition remained near the midpoint of the morph sequence across all five sequences; in one mouse it began near the midpoint of the first morph sequence, but gradually shifted over time (Fig. 1e), similar to a previous report[8]. In all cases, the familiar and morph environments remained partially correlated with one another, though final morph sequences were significantly more decorrelated than the first morph sequence (Fig. 1f). This increased decorrelation might be a result of ongoing learning from experience with the morph sequence itself and/or the daily unrecorded top-up experience in each of the familiar environments. Altogether, these results demonstrate that our adapted geometric morph paradigm replicates many phenomena observed within-sequence in acute versions of this paradigm and suggests that these dynamics may continue to evolve with experience.

**Context and drift are distinct at the population level**. Next, we explored how population-level representational similarity

changed across the 32 days of recording. Because of variability in whether the same cell met traditional within-session place cell criteria across sessions (Fig. S5), and to avoid any biases that subselecting cells based on functional tuning properties might introduce, all cells were eligible for inclusion regardless of their spatial tuning during a given session. Furthermore, to ensure that our results were not driven by unequal numbers of cells tracked across session comparisons, we subsampled our data to match the minimum number of tracked cells across all pairwise comparisons for each mouse.

With these details in mind, we first computed the mean rate map similarity across equal numbers of tracked cells for each pairwise comparison of sessions (Fig. 2a). These matrices exhibited structural features which generalized across mice: mean rate map correlations vary from high values for pairs of sessions close in time toward zero for sessions pairs separated further in time, with additional modulation by context. To make this structure explicit, we reduced this similarity matrix to a two-dimensional embedding via nonmetric multidimensional scaling (nMDS; see Methods)[25]. MDS is a set of unsupervised techniques for transforming a potentially high-dimensional pairwise distance matrix into a set of points in a low-dimensional space while preserving the relative distances between points as well as

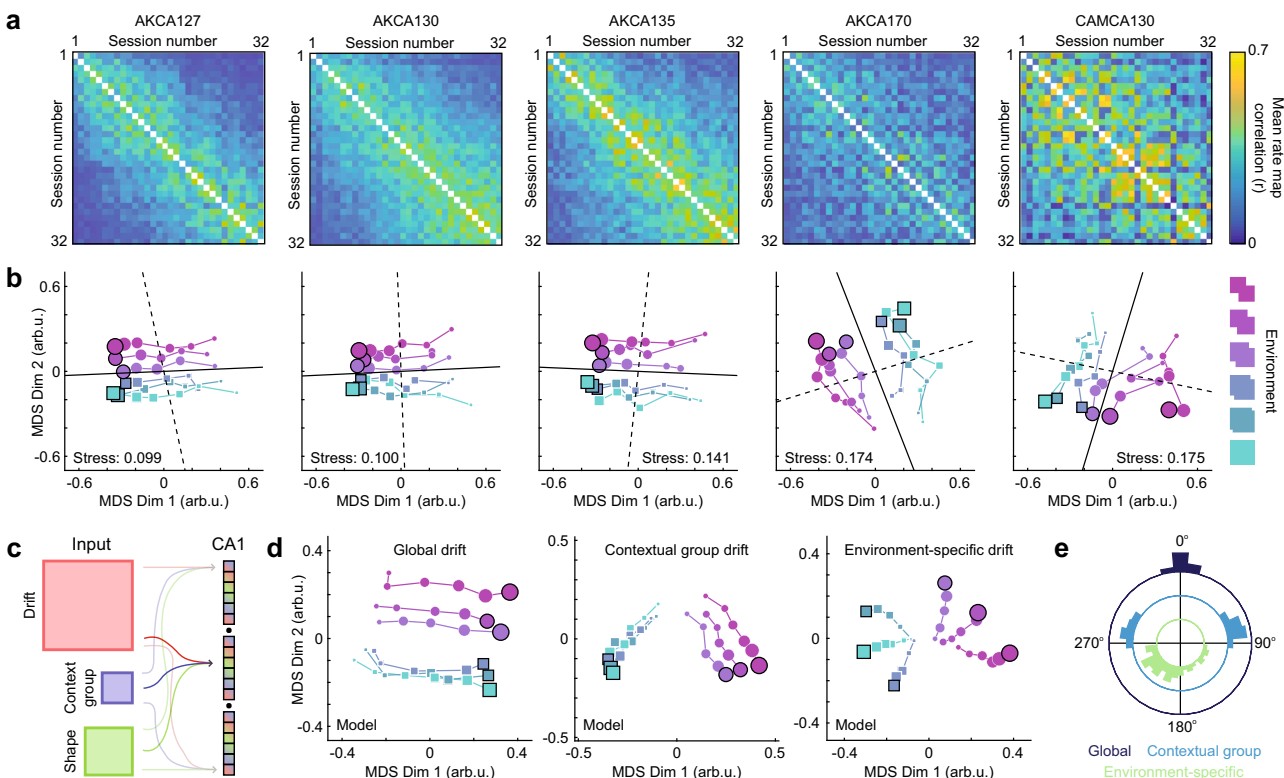

**Fig. 2 Nonmetric multidimensional scaling reveals distinct temporal and contextual components governing the CA1 spatial code. a** Population-level representational similarity matrices across all recording sessions for all five mice. Each RSM is computed by taking the mean rate map correlation across all tracked cells for each pairwise comparison of sessions. Mouse name at the top of each column. **b** Session similarity structure when embedded in a two-dimensional space via nonmetric multidimensional scaling (nMDS). Dot size indicates session number, with earliest sessions indicated by the smallest dots and latest sessions indicated by the largest dots. The final sessions are outlined in black. Stress values, a measure of goodness-of-fit, were below 0.2 in all cases indicating sufficiently faithful embeddings[54]. Note that the estimated drift dimension (solid black line) and context dimension (dashed black line) are nearly orthogonal in this embedding (see Methods). **c** CA1 populations were modeled as a combination of spatial inputs which were modulated by drift, contextual group, and the shape of the environment. **d** Example nMDS embeddings of modeled populations for different drift dynamics. **e** Distributions of the angular difference between the direction of drift estimated from both familiar environments for global, contextual group, and environment-specific drift dynamics. "arb.u." denotes arbitrary units.

possible, as defined by some cost function. In the case of nonmetric MDS, the cost function aims to preserve the rank order relationships between points but tolerates a nonparametric monotonic scaling of distances, which is useful when measured distances are not expected to scale linearly as is the case with correlation distances. A strength of MDS is its applicability to cases where the relative similarity between conditions is measured, not the raw position of each condition in a common feature space, as is the case here. Thus, in the current application nMDS will map the mean pairwise rate map similarity matrix to a two-dimensional embedding where each session is a point and the relative position of sessions indicates their similarity, while the raw scale and overall orientation of the arrangement are arbitrary.

The resultant embeddings revealed two dimensions that strongly determined representational similarity in this paradigm: a contextual component and a drift component (Fig. 2b). These components were pronounced in every mouse. Quantification of the embedded representations revealed that the contextual and drift components defined nearly orthogonal dimensions in this 2D subspace, with absolute angular differences between context and drift dimensions centered around 90° (see *Methods*; all angles: [84.7°, 86.4°, 89.4°, 94.1°, 99.7°]; Rayleigh's test versus uniformity on the 0° to 180° range: $p = 2.4e{-3}$, $z = 4.82$). Estimations of the direction of drift from repeated visits across time to both familiar environments were similar to one another

(absolute angular difference between drift direction estimates: [4.4°, 13.9°, 20.5°, 22.4°, 24.1°]). A qualitatively similar distinction between contextual and drift components was observed when embedding in three dimensions (Fig. S6) and when using population vector correlations as the measure of similarity between pairwise session comparisons (Fig. S7).

To situate these findings, we constructed a computational model in which individual CA1 cells were simulated as the rectified sum of spatially-tuned inputs modulated by drift, contextual group (grouping together each half of the shapespace), and the shape of the environment (Fig. 2c; see Methods). Next, we varied the dynamics of the drift input, such that drift accrued dependent on the specific environment, the context group, or simply as a function of time (global drift). Each drift dynamic resulted in a different stereotyped nMDS embedding (Fig. 2d). Global drift yielded an embedding in which drift and contextual components were orthogonal, and the direction of drift estimated from both familiar environments were consistent (Fig. 2e). Context group-dependent drift also resulted in roughly orthogonal drift and context components; however, estimates of the direction of drift differed between the two familiar environments. Environment-specific drift yielded a qualitatively distinct inside-outward radial embedding in which each environment became more distinct over time, leading to roughly opposing estimates of the direction of drift between the familiar environments. Thus, only global drift reproduced essential characteristics of

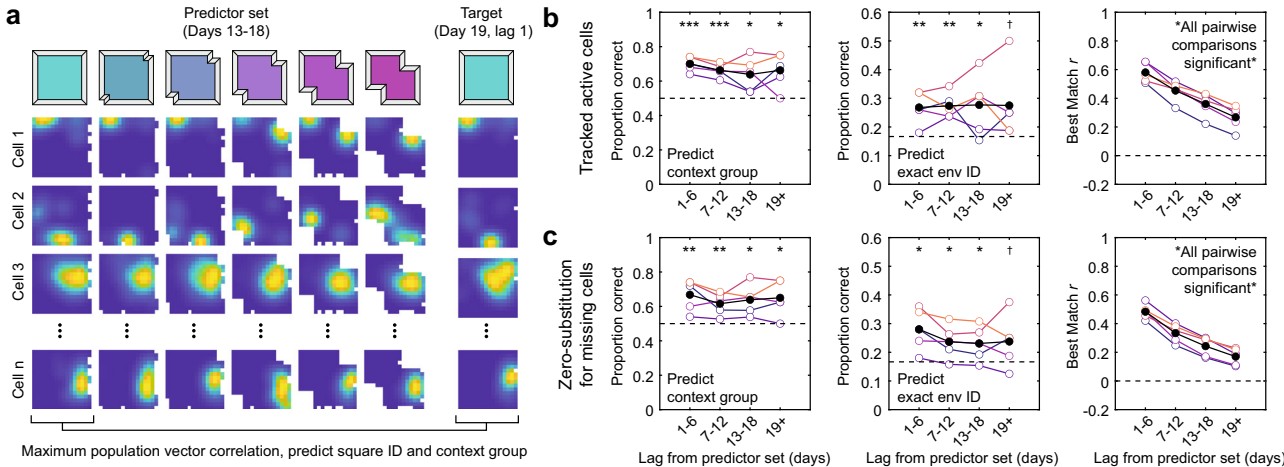

**Fig. 3 An accurate readout of context information is preserved across weeks despite neural drift. a** To predict contextual group and exact environment identity, each target session was compared via population vector correlation to each environment in a six-day contiguous predictor set morph sequence. The contextual properties associated with the highest population vector correlation to the target were then predicted for the target session. This process was repeated for each target day using each six-day morph sequence as the predictor set, and results were aggregated across predictor sets and lags. Rate maps normalized from zero (blue) to the peak (yellow) within each session. **b** Proportion correct for contextual group prediction (left), proportion correct for exact environment ID prediction (middle), and best matching population vector correlation (right) as a function of lag between the predictor set and the target session, including all active cells tracked between each pair of comparisons. All five individual mice are color-coded as in Fig. 1, mean in black. **c** As in (**b**) except for all cells while substituting all-zero rate maps for cells which were inactive/not identified on a given session. Statistical markers denote the outcomes of a one-tailed *t*-test versus chance accuracy, exact statistical outcomes are reported in Supplementary Table 1. Source data provided as Source Data file. $^{\dagger}p < 0.10$, $^{*}p < 0.05$, $^{**}p < 0.01$, $^{***}p < 0.001$.

the representational structure of the CA1 spatial code in this paradigm.

**Accurate contextual information can be read out across weeks.** If neural drift preserves the relative structure of the representation of context, then we should be able to accurately predict context information across time, even on the basis of a temporally-contiguous subset of the data. To address this possibility, we attempted to predict both context group identity (grouping together each half of the shapespace) and the specific environment identity on the basis of population vector rate map similarity to each session in a contiguous six-day predictor morph sequence (Fig. 3a). Specifically, we took the sessions from a given six-day morph sequence and a target session outside of the predictor set and predicted the context group and exact environment identity from the best matching predictor set session. Then we repeated this process for all target sessions and using each 6-day morph sequence as the predictor set once, and aggregated the results according to the lag in days between the target session and the nearest predictor set session. We conducted this analysis while treating inactive/unidentified cells in two different ways. In one case, we included only the active cells which were tracked between each pair of comparisons when computing the population vector correlation. In another case, we included all cells, substituting zeros for the cases where cells were inactive/not identified on a given session.

In all cases, prediction accuracy remained high even with 19+ days (>3 sequences) separating the predictor sequence and target day, with ANOVAs indicating no significant differences between epochs (Fig. 3b, c). Notably, prediction accuracy remained high even though the similarity between the target session and the best matching predictor set session significantly decreased as a function of lag, reflecting the ongoing drift (Fig. 3b, c). Together, these results provide additional evidence that drift preserves the relative representational structure of context, allowing accurate readout of context information across a weeks-long timescale.

**Heterogeneous single-cell selectivity for context and drift.** The preceding results indicate that the differences in neural activity which distinguished context were distinct from the differences in neural activity which accompanied the passage of time at the level of the population. This structure might arise from varying degrees of selectivity at the level of individual cells. To address these possibilities, we next examined the relationship between context coding and long-term stability at the level of individual cells. We did so in two ways. First, we took a generalized linear modeling (GLM) approach. For each cell, we computed the pattern of similarity between all pairs of sessions for which that cell was identified, with similarity quantified by the rate map correlation, which we termed the representational similarity matrix (RSM) for that cell. Then, for each cell with a sufficient number of comparisons (identified on at least 16/32 sessions; 120 pairwise comparisons), we attempted to explain the variance in its RSM via a GLM as a function of three factors (and their interactions): drift, context group, and shape (Fig. 4a). The drift factor specified a linear decrease in similarity as a function of the number of days between each pairwise session comparison. The context group factor specified high similarity between sessions from the same side of the shapespace and low similarity otherwise, with the within-sequence transition point determining the cut-off between sides of the shapespace (i.e., Fig. 1e). The shape factor specified a linear decrease in similarity as a function of the distance in the shapespace between each session pair. These three factors (and their interactions) provided an effective characterization at the level of the population, explaining the majority of the variance of the population RSM (i.e., mean RSM across cells) for all animals [full model $r^2$: 0.747, 0.787, 0.816, 0.835, 0.894].

Applying this full model to a given cell RSM provided a quantification of how much RSM variance could be explained by all three factors. Next, we sought to isolate the contributions of separate factors to this full model explained variance. To do so, we dropped individual factors (and interactions with those factors) from the model, recomputed the resulting explained variance, and compared the reduction in explained variance

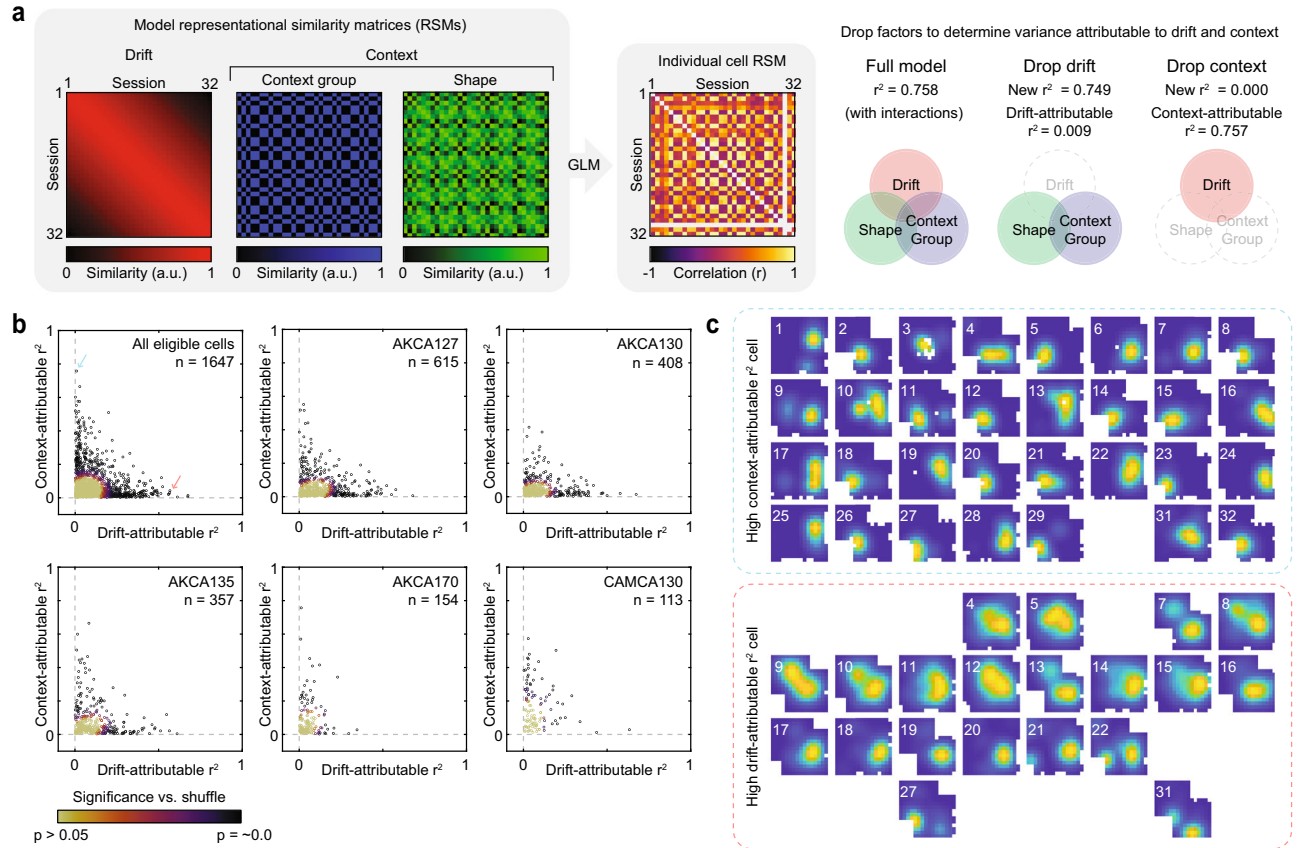

**Fig. 4 Heterogeneity in the representational similarity of sessions across contexts and time at the level of individual cells. a** Schematic of the full model used to explain individual cell RSM variance via GLM (left), and the procedure of dropping individual factors and their interactions to determine attributable explained variance ($r^2$; right). **b** Distribution of explained variance attributable to drift and contextual factors for all cells identified on at least 16 sessions, combined and separated by an animal. The color indicates uncorrected significance relative to a nonparametric shuffled control distribution as described in the main text. **c** Example rate maps for all identified sessions for two example cells, one high in context-attributable explained variance (top; blue border) and one high in drift-attributable explained variance (bottom; red border) arranged in chronological order. The position of these example cells in the combined distribution in (**b**) is indicated by arrows of the matching color. Session number is located in the top-left corner of each rate map. Rate maps normalized from zero (blue) to the peak (yellow) within each session. Source data provided as Source Data file.

against the full model. This reduction provided a quantification of the RSM variance attributable to a given factor. For ease of interpretation, we removed both context group and shape together when dropping factors to estimate their combined contribution, referring to the outcome as a context-attributable variance.

This analysis revealed significant heterogeneity in the factors which explained RSM variance at the level of individual cells (Fig. 4b). For many cells, RSM variance was explained by either the drift factor or the contextual factors, but not both (Fig. 4c). To situate these results, we compared the distribution of attributable explained variance to a shuffled distribution, in which we shuffled the RSM values across eligible cells independently for each pairwise comparison of sessions. This shuffled distribution preserves the population RSM and estimates the distribution of variance attributable to drift and contextual factors that one would expect on the basis of the variability within the population alone. For each cell, we then computed the Mahalanobis distance from this shuffled distribution, yielding a measure of significance. A large portion of cells exhibited patterns of attributable explained variance which deviated significantly from this shuffled distribution ($p < 0.01$, $n = 453$ of 1647, 27.5%; binomial test, $p = \sim0.0$), with many of these cells primarily loading onto either drift or context factors, but not both. Similar results were

observed when varying inclusion criteria and the details of the model (Fig. S8).

Our GLM approach revealed evidence of heterogeneous RSM patterns at the level of individual cells, isolating cells with strongly context-loading patterns with little evidence of drift, and vice versa. In our second approach, we sought to address this potential relationship directly. To do so, we must quantify both contextual coding and long-term stability using distinct subsets of the data to ensure that any relationship we observe is not the product of biases induced by resampling the same data.

With this in mind, we first characterized the extent to which the pattern of activity for a given cell during a given six-day morph sequence resembled an interpretable contextual code. To operationalize interpretable contextual coding in a way which tolerates the heterogeneous but stereotyped responses we observed between cells in our within-sequence analysis (i.e., Fig. 1c), for each cell we first computed its within-sequence contextual RSM, a matrix which summarizes the similarity of each environment to every other environment for a given morph sequence (Fig. 5a and Fig. S9). Next, we fit this contextual RSM with a five-parameter sigmoidal model that could capture a wide range of interpretable dynamics (Fig. 5b). Finally, we took the (inverse) mean squared error of this model fit as our measure of interpretable contextual coding. As our measure of long-term

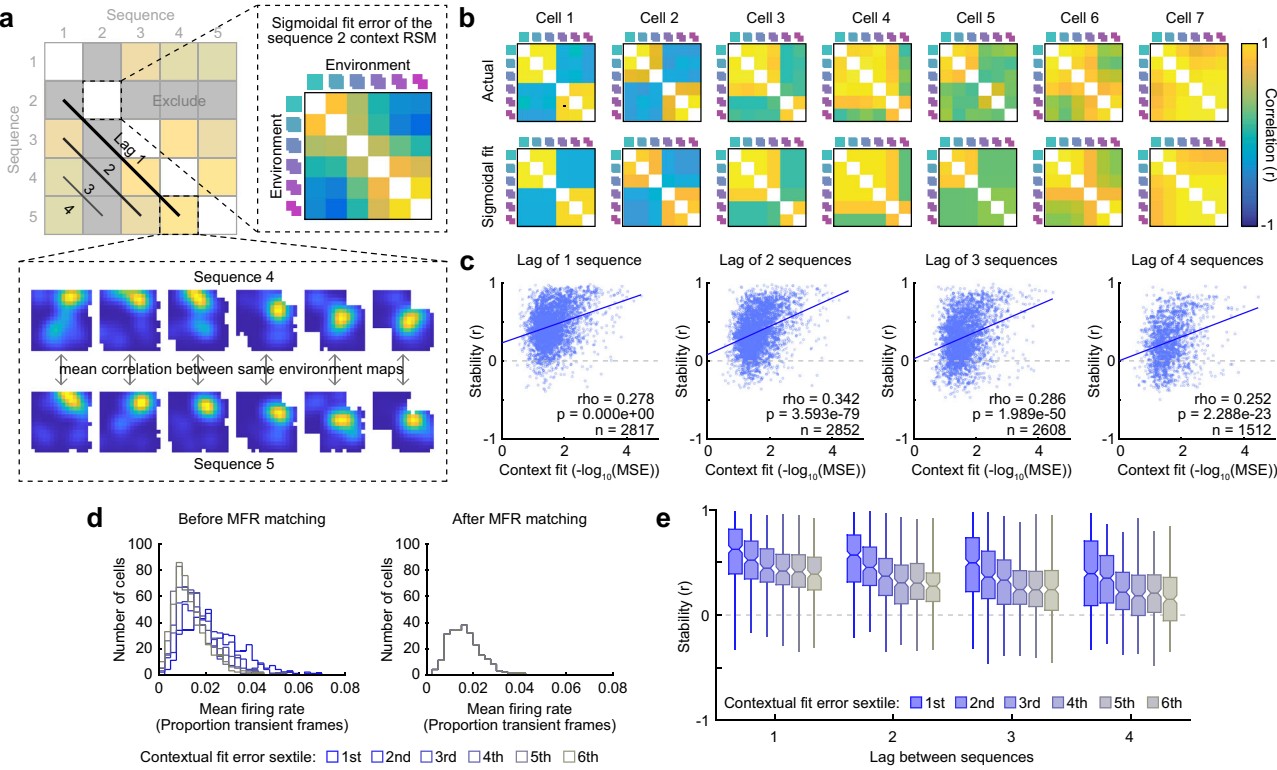

**Fig. 5 Interpretable contextual coding is associated with long-term stability at the level of individual cells. a** Schematic for quantifying the relationship between contextual coding and long-term stability at the single-cell level. **b** Seven examples of measured contextual RSMs for a given morph sequence, and their five-parameter sigmoidal fits. **c** The relationship between contextual RSM goodness-of-fit and long-term stability at the single-cell level, aggregated by the lag between the pairs of sequences from which long-term stability was measured. The solid line indicates the least-squares fit. The outcome of Spearman's rank correlation was noted. **d** Example of mean firing rate histograms for cells in each contextual fit error sextile before and after histogram matching by subsampling cells in each group. **e** Long-term stability at each lag for cells in each contextual fit error sextile, after subsampling cells within lag in each sextile to match mean firing rate histograms. At all lags, cells in the lower sextiles (i.e., cells with greater contextual goodness-of-fit) were significantly more stable across sequences than cells in other sextiles, even after controlling for differences in mean firing rates. ANOVAs at all lags are significant. Exact statistical outcomes and uncorrected two-tailed two-sample *t*-test post hoc contrasts are reported in Supplementary Table 2. Box plots portray the minimum and maximum (whiskers), central quartiles (boxes), and median (cinch). Source data provided as Source Data file.

stability, we computed the mean rate map correlation for the same environments across each pair of six-day morph sequences, excluding the morph sequence used to quantify the extent of interpretable contextual coding (Fig. 5a). Note that this measure of stability assumes nothing about the relative similarity across different environments and only indexes the similarity of the representation of the same environment across time. We repeated this process using each 6-day morph sequence for the contextual fit once and combined the results, averaging across equivalent lags between sequences when estimating long-term stability. Across all lags, these measures of context coding and stability were correlated (Fig. 5c), indicating that cells with a more interpretable contextual RSM during a given sequence tended to have more stable rate maps across withheld sequences, even when withheld sequences were separated by weeks.

Notably, however, both of these measures were correlated with firing rate: cells which had higher firing rates tended to have lower contextual fit errors (Spearman's rank correlation: $\rho = -0.358$, $p = \sim 0.0$, $n = 3042$) and higher rate map stability across sequences (Spearman's rank correlation: Lag of 1 sequence: $\rho = 0.190$, $p = 3.347e-24$, $n = 2817$; Lag of 2 sequences: $\rho = 0.209$, $p = 2.040e-29$, $n = 2852$; Lag of 3 sequences: $\rho = 0.160$, $p = 1.788e-16$, $n = 2608$; Lag of 4 sequences: $\rho = 0.178$, $p = 3.103e-12$, $n = 1512$). To mitigate the influence of this potential confound when assessing the relationship between contextual coding and long-term stability, we took the following approach. First, we again computed the contextual

RSM fit error for a given 6-day morph sequence for each cell tracked across this sequence. Next, we divided these cells into six groups according to the sextile of their contextual fit error. We then randomly subsampled the cells in each of these six sextile groups to match the mean firing rate distributions during this contextual fit sequence across all six groups (Fig. 5d). Finally, we computed the stability between all remaining pairwise comparisons of morph sequences for each group of subsampled cells, excluding the contextual fit sequence. We repeated this process using each 6-day morph sequence for the contextual fit once and aggregated the results across similar lags. This analysis revealed that cells with a more interpretable contextual RSM on a given sequence tended to have more stable rate maps across withheld sequences, even when controlling for covarying firing rate differences (Fig. 5e).

## Discussion

Here we recorded from CA1 in freely behaving mice over 32 days of experience in an adapted geometric morph paradigm. On a shorter within-sequence timescale, the hippocampal representation resembled prior acute reports[8,9,26], exhibiting sigmoidal population-level contextual similarity dynamics across the shapespace which were driven by heterogeneous but stereotyped single-cell patterns of activity. Characterizing the representational structure across all 32 days revealed that changes indicative of context were orthogonal in network space to changes which

accompanied the passage of time. This specific structure was well described by a global drift model whereby drift is accrued independently of context identity. This structure enabled consistent readout of contextual information across a timescale of weeks even on the basis of a temporally-contiguous predictor set, suggesting that downstream contextual readout tuned to the representation at a particular time can successfully generalize across long timescales. Lastly, individual cells exhibited a relationship between interpretable contextual coding and long-term stability even when controlling for covarying differences in firing rate, suggesting that propensity to drift is heterogeneous at the level of individual cells and possibly linked to functional content. Together, these results demonstrate that the relative structure of the hippocampal representation of context is preserved despite ongoing changes to the spatial code on a timescale of weeks.

Recent results have provoked claims of representational drift at unexpected rates not only in the hippocampus[13–16] but also in cortical sensorimotor regions[19,27,28]. In each of these cases, a claim of drift is made on the basis of critical assumptions about the form and content of the representation, specifically that the content of the representation is known and that the way this representation is encoded is also known. For example, for claims of drift within the hippocampus, it is implicit in the analysis that the content of the representation is spatial and that the form of this representation is a population-wide rate code. While each of these assumptions are well motivated by prior work, there are alternative views that might accommodate the observed long-timescale dynamics without a claim of representational drift. Mounting cross-species experimental[29–33] and theoretical[34–39] work suggests that the hippocampal representation is fundamentally superspatial and includes a temporally-correlated dimension, either implicitly[36,40] or explicitly[34,37,38]. Moreover, it is known that the activity of hippocampal neurons is remarkably structured and coordinated on short timescales[40–43], suggesting that a rate code (spatially-conditioned or not) may provide only a partial assay of representational content in this region. Because of these alternatives, the degree to which long-timescale changes to the hippocampal code reflect representational content versus drift remains open to debate, perhaps more so than in sensory regions where the coded content might be more completely understood.

The results we present here speak to this debate. Characterization of the population-level representational structure via dimensionality reduction revealed that changes in network activity which were indicative of spatial context were orthogonal to changes accompanying the passage of time, thus preserving the relative structure of contextual representation. While the changes accompanying the passage of time could reflect global drift independent of spatial context, this component could alternatively reflect a faithful reproduction of representational content which also varied on this timescale, for example representations of continuous experiences, ongoing learning, predictive relationships, or time itself. Consistent with this, across two different analytical approaches we observed corroborative evidence of a relationship between interpretable contextual coding and long-term stability, even when controlling for known firing rate covariates[44]. This link suggests that long-term instability in spatial tuning properties might be driven by the representation of content beyond the spatial context. CA1 is known for genetic[45], anatomical, and functional heterogeneity[46–48] in the responses of individual cells, which is linked to biases in upstream inputs[49,50]. In addition to differences in their coding properties and content, these inputs also differ radically in their long-term stability[15,51], providing a potential link between representational content and long-term dynamics within CA1. This possibility thus motivates future work dissecting circuit-specific contributions to the representation of specific content and accompanying long-term dynamics.

Currently, it is unknown whether and how long-timescale changes attributed to drift are coordinated across the brain. One possibility is that drift is coordinated in such a way that incongruences between regions are continuously corrected[19]. On the other hand, here we show that information about the spatial context can be predicted with high accuracy across weeks despite ongoing representational changes, even on the basis of temporally-contiguous training data. This suggests that a downstream readout of contextual information trained at a particular moment in time will continue to generalize well on long timescales, even in the absence of continued coordination between the hippocampus and this hypothetical reader.

Here we leveraged the partial correlation between hippocampal maps of similar spaces to characterize the relationship between representational content and long-timescale dynamics. While necessary for this characterization, we are therefore limited in our ability to speak to ongoing changes in other portions of the representational statespace. Thus, while our results suggest that a global model in which long-timescale changes accrued independently of spatial context is most appropriate, we cannot rule out globally-inconsistent dynamics in other portions of the representational space.

In sum, our results demonstrate that the relative structure of contextual representation in CA1 is preserved despite continuous representational changes on the timescale of weeks. These results speak to an ongoing debate about representational content and drift throughout the brain, with fundamental implications for the nature of hippocampal coding. Finally, these results motivate future experiments dissecting the relationship between functional properties and long-term dynamics within and beyond the hippocampal formation.

## Methods

**Subjects**. Naive male mice (C57Bl/6, Charles River) were housed in pairs on a 12-h light/dark cycle at 22 °C and 40% humidity with food and water ad libitum. All experiments were carried out during the light portion of the light/dark cycle, and in accordance with McGill University and Douglas Hospital Research Centre Animal Use and Care Committee (protocol #20157725) and with Canadian Institutes of Health Research guidelines.

**Surgeries**. During all surgeries mice were anesthetized via inhalation of a combination of oxygen and 5% Isoflurane before being transferred to the stereotaxic frame (David Kopf Instruments), where anesthesia was maintained via inhalation of oxygen and 0.5–2.5% Isoflurane for the duration of the surgery. Body temperature was maintained with a heating pad and eyes were hydrated with gel (Optixcare). Carprofen (10 ml kg$^{-1}$) and saline (0.5 ml) were administered subcutaneously at the beginning of each surgery. Preparation for recordings involved three surgeries per mouse.

First, at the age of 6 to 10 weeks, each mouse was transfected with a 350 nl injection of the calcium reporter GCaMP6f via the viral construct AAV5.CaMKII.GCaMP6f.WPRE.SV40 (CAMCA130) or AAV9.syn.GCaMP6f.WPRE.SV40 (all other mice). The original titer of the AAV9.syn.GCaMP6f.WPRE.SV40 construct, sourced from University of Pennsylvania Vector Core, was 3.26e14 GC-ml and was diluted in sterile PBS (1:1 AKCA170; 1:5 AKCA127; 1:10 AKCA130; 1:15 AKCA135) before surgical microinjection. The original titer of the AAV5.CaMKII.GCaMP6f.WPRE.SV40 construct, sourced from Addgene, was 2.3e13 GC-ml and was diluted in sterile PBS (1:3) before surgical microinjection.

One to 3 weeks post-injection, either a 1.8 mm (AKCA127, AKCA130, AKCA135) or 0.5 mm (AKCA170, CAMCA130) diameter gradient refractive index (GRIN) lens (Go!Foton) was implanted above the dorsal CA1 (Referenced to bregma: ML = 2.0 mm, AP = −2.1 mm; Referenced to brain surface: DV = −1.35 mm). Implantation of the 1.8 mm diameter GRIN lens required aspiration of intervening cortical tissue, while no aspiration was required for implantation of the 0.5 mm diameter GRIN lens. Results observed using 1.8- or 0.5-mm diameter GRIN lenses were similar. In addition to the GRIN lens, two stainless steel screws were threaded into the skull above the contralateral hippocampus and prefrontal cortex to stabilize the implant. Dental cement (C&B Metabond) was applied to secure the GRIN lens and anchor screws to the skull. A silicone adhesive (Kwik-Sil, World Precision Instruments) was applied to protect the top surface of the GRIN lens until the next surgery.

One to 3 weeks after lens implantation, an aluminum baseplate was affixed via dental cement (C&B Metabond) to the skull of the mouse, which would later secure the miniaturized fluorescent endoscope (miniscope) in place during recording. The miniscope/baseplate was mounted to a stereotaxic arm for lowering above the implanted GRIN lens until the field of view contained visible cell segments and dental cement was applied to affix the baseplate to the skull. A polyoxymethylene cap with a metal nut weighing ~3 g was affixed to the baseplate when the mice were not being recorded, to protect the baseplate and lens, as well as to simulate the weight of the miniscope.

After surgery, animals were continuously monitored until they recovered. For the initial 3 days after surgery mice were provided with a soft diet supplemented with Carprofen for pain management (MediGel CPF, ~5 mg kg$^{-1}$ each day). Familiarization with both environments (while recording in the room A-associated environment to monitor imaging quality and habituate the mouse to recording) began 3 to 7 days following the baseplate surgery.

**Data acquisition**. In vivo calcium videos were recorded with a UCLA miniscope (v3; miniscope.org) containing a monochrome CMOS imaging sensor (MT9V032C12STM, ON Semiconductor) connected to a custom data acquisition (DAQ) box (miniscope.org) with a lightweight, flexible coaxial cable. The DAQ was connected to a PC with a USB 3.0 SuperSpeed cable and controlled with Miniscope custom acquisition software (miniscope.org). The outgoing excitation LED was set to between 2–8% (~0.05–0.2 mW), depending on the mouse to maximize signal quality with the minimum possible excitation light to mitigate the risk of photo-bleaching. The gain was adjusted to match the dynamic range of the recorded video to the fluctuations of the calcium signal for each recording to avoid saturation. Behavioral video data were recorded by a webcam mounted above the environment. Behavioral video recording parameters were adjusted such that only the red LED on the CMOS of the miniscope was visible. The DAQ simultaneously acquired behavioral and cellular imaging streams at 30 Hz as uncompressed avi files and all recorded frames were timestamped for post hoc alignment.

All recording environments were constructed of a gray Lego base and black Lego bricks (Lego, Inc). The full square environment was 38 cm × 38 cm. All walls had a height of 22 cm. During recording, the environment was dimly lit by a nearby computer screen, which could serve as a directional cue. During familiarization and all unrecorded experiences, the environments were well-lit by overhead room lighting. All sessions were 20 min, and only one session was recorded per day to avoid photobleaching. Following the recorded session, mice were returned to their home cage for 5 min, after which the unrecorded top-up experience in the familiar environments began. Mice were directly transported from one familiar environment to the other between unrecorded sessions. Each familiar environment session took place in a different neighboring room (e.g., room A:square environment, and room B:other familiar environments), with an assignment of each familiar environment to a given room kept constant within each mouse and randomized across mice. All recordings took place in one of these rooms (e.g., room A). The two familiar environments were always recorded in the same order (e.g., square day $n$, other familiar environment day $n + 1$) within each mouse, with the order randomized across mice. The order of the four morph environments was randomized for each sequence. The mouse was always placed in the same corner at the start of the session and was allowed to explore the environment for 15 to 30 s prior to the start of data acquisition. Following each session, the environment was cleaned with disinfectant (Prevail).

**Data preprocessing**. Calcium imaging data were preprocessed prior to analyses via a pipeline of open-source MATLAB (MathWorks; version R2021a) functions to correct for motion artifacts[20], segment cells and extract transients[21,22]. To extract the rising phase of transients from each filtered calcium trace, we proceeded as follows. First, we computed the derivative of the calcium signal, smoothed with a gaussian kernel with a standard deviation of 5 frames. Next, because calcium transients around the baseline can only be positive, we estimated the variance in the derivative of the smoothed calcium signal on the basis of noise via a half-normal distribution such that:

$$\text{NOISE} = \frac{\text{std}(\triangle \mathbf{t}(\mathbf{t} < 0))}{\sqrt{1 - \frac{2}{\pi}}} \qquad (1)$$

where $\triangle t$ is the smoothed time-derivative of the median-subtracted calcium trace $\mathbf{t}$. We then $z$-scored $\triangle \mathbf{t}$ on the basis of this noise distribution. The final binarized rising-phase vector was then set to 1 whenever this $z$-scored $\triangle \mathbf{t}$ vector exceeded 2.5, and 0 otherwise. This binary vector was treated as the firing rate in all further analyses. The motion-corrected calcium imaging data were manually inspected to ensure that motion correction was effective and did not introduce additional artifacts. Following this preprocessing pipeline, the spatial footprints of all cells were manually verified to remove lens artifacts. Position data were inferred from the onboard miniscope red LED offline following recording using a custom-written MATLAB (MathWorks) script and were manually corrected if needed. Cells were tracked across sessions on the basis of prominent landmarks, their spatial footprints, and/or centroids[23].

**Data analysis**. All analyses were conducted using the binary vector of the rising phases of transients, treating this vector as if it were the firing rate of the cell

(henceforth firing rate). Similar results were observed when the likelihood of spiking was inferred via a second-order autoregressive deconvolution[52], instead of transient rising-phase extraction.

Rate maps were constructed by first binning the position data into pixels corresponding to a 2.5 cm × 2.5 cm grid of locations. Then, to correct for biases in sampled spatial locations, we subsampled our data during all rate map comparisons to match the spatial sampling distributions across comparisons. To do so, we computed the minimum number of samples recorded at each pixel location across comparisons. Next for each comparison, we included a random subset of the data recorded at each pixel location to match that minimum number of samples. On the basis of these subsampled data, we next computed the mean firing rate for each pixel and then smoothed this map with a 4 cm standard deviation isometric Gaussian kernel. For comparisons between rate maps, the similarity was measured as the Pearson's correlation between corresponding pixels.

Within-sequence analyses summarized morph sequence dynamics with transition plots (Fig. 1d and Fig. S4), which captured the similarity of all six morph environment maps to the two familiar environment maps. To this end, rate map correlations between each environment and familiar rate maps at the beginning and end of the morph sequence were computed. Only comparisons between cells whose within-session split-half rate map correlation (SHC) exceed the 95th percentile of a shuffled control for at least one of the compared sessions were included. The shuffled control was computed for each cell by randomly circularly shifting its firing rate vector relative to the position data by at least 30 s and recomputing the SHC 1000 times to create the null distribution. To characterize these plots, the Fisher-transformed median values comparing the morph environments to each of the two familiar environments were both fit to minimize mean squared error with a four-parameter sigmoid of the form

$$f(x) = p_3 + \left(\frac{1}{1 + e^{((-x + p_1)p_2)}}\right) / p_4 \qquad (2)$$

where $x$ is the position of the current environment in the shapespace (arbitrarily chosen to range from 1 to 6), and parameters $p$ determine the shape of the sigmoid. The intersection of these two sigmoids was taken as the measure of the transition point between familiar maps. The maximum absolute difference between these two sigmoids within the sampled shapespace was taken as the measure of maximum map decorrelation.

Embeddings of the population representation via nonmetric multidimensional scaling were computed and quantified as follows. First, we computed the mean rate map correlation across cells for each pairwise comparison of sections, subsampling to match the spatial sampling distributions between comparisons and so that all comparisons included only the minimum number of cells tracked across all pairwise comparisons, as described in the main text. Next, we transformed this correlation matrix into a distance matrix by computing one minus this correlation matrix, with the diagonal set to a distance of 0. Finally, this distance matrix was reduced to a two-dimensional embedding via nonmetric multidimensional scaling[25], implemented by the MATLAB function mdscale with the default parameterizations and minimizing Kruskal's normalized stress1 cost function:

$$\text{stress1} = \sqrt{\frac{\sum_{i=1}^{n}\sum_{j=i+1}^{n}\left(\text{F}\left(d_{ij}\right) - d_{ij}\right)^2}{\sum_{i=1}^{n}\sum_{j=i+1}^{n}d_{ij}^{2}}} \qquad (3)$$

where $d_{ij}$ is the distance between $i$ and $j$ in the measured distance matrix and $\text{F}(d_{ij})$ is a nonparametric monotonic function of the measured distances fit via isotonic regression. To estimate the context dimension within two-dimensional embeddings, the average difference separating neighboring-in-time familiar environment recordings was computed. To estimate the time dimension within two-dimensional embedding, the average difference between the first and last recordings of each familiar environment was computed. In Supplementary Figures, this analysis was expanded to include three-dimensional embeddings and to take the population vector correlation as the measure of similarity between sessions.

Individual cell contextual representational similarity matrices were computed for a given cell and sequence as the Pearson's correlation between rate maps for all pairwise comparisons between the six environments (after subsampling to match spatial sampling distributions), resulting in a 6 × 6 matrix. To characterize individual cell contextual RSMs in various ways, each two-dimensional RSM was fit with a five-parameter sigmoidal model as described by these two equations:

$$s(x) = \left(\frac{1}{1 + e^{((-x + p_1)p_2)}}\right) + p_3 \qquad (4)$$

$$S(x, y) = (s(x) \oplus s(x)^{\text{T}}) / p_4 + p_5 \qquad (5)$$

here $s(x)$ described a one-dimensional sigmoidal function evaluated at $x$ (in this case x = [1, 2, 3, …, 6] to span the six environments), $s(x)^{\text{T}}$ denotes the transpose of this vector, $\oplus$ denotes the pairwise element multiplication of these two vectors [MATLAB's *bsxfun(@times*,a,b)], and $p$ is the vector of five free parameters to be fit by reducing the mean squared error between $S(x,y)$ and the target contextual RSM for the upper triangle of each, as contextual RSMs are symmetric across the diagonal. These parameters have intuitive interpretations: $p_1$ determines the transition point for map similarity in the morph sequence, $p_2$ determines the

abruptness of the transition, $p_3$ determines the asymmetry in the similarity between halves of the shapespace, $p_4$ determines the degree of sigmoidal modulation, and $p_5$ determines the average overall similarity of the RSM. The goodness-of-fit of this model was taken as a measure of the extent to which a given contextual RSM exhibits interpretable contextual coding, as this model is capable of flexibly accounting for a wide range of interpretable dynamics.

Spatial information content (SIC) was computed from the whole-session rate maps of each cell as described previously[53] via the equation:

$$\mathrm{SIC} = \sum_i \mathbf{s}_i \mathbf{r}_i \log\left(\frac{\mathbf{r}_i}{\bar{\mathbf{r}}}\right) \qquad (6)$$

where $i$ is the rate map pixel index, $\mathbf{s}_i$ is the probability of sampling pixel $i$, $\mathbf{r}_i$ is the mean firing rate at pixel $i$, and $\bar{\mathbf{r}}$ is the mean firing rate across all pixels.

**Modeling CA1 drift dynamics**. To simulate the representational structure of various drift dynamics, we created a rate map-based model. For ease of modeling, rate map binning was set in Lego coordinates, with the environment being 48 Lego pips wide (approximately equivalent to 38 cm).

First, we created three populations of $48 \times 48$ pixel input rate maps which varied in their dynamics: a shape population (150 cells), a contextual group population (25 cells), and a drift population (300 cells). For shape inputs, a single-pixel was selected as the field center for each cell. Next, for all sessions besides the square, if this pixel was in a row/column that was affected by the deformation then its location was rescaled to match the deformation. Half of the geometry inputs were sensitive to the x-axis during deformations, with the other half sensitive to the y-axis during deformations. For contextual group inputs, the three most square-like environments were assigned a single random pixel as the field center for those sessions. A different random pixel was then assigned for the remaining half of the shapespace.

Drift input rate maps were created differently depending on the modeled drift dynamics. In the case of global drift, a single-pixel was randomly selected as the field center for each cell. Over consecutive days, this pixel was continuously shifted by a random amount on the range [−4, 4], inclusive, independently on both axes. For contextual group-specific drift dynamics, this drift accrued independently for environments in each half of the shapespace, such that over time each contextual group grew more dissimilar. For environment-specific drift dynamics, this drift accrued independently for every environment, leading all environments to become more dissimilar from one another over time. All input rate maps were then smoothed with an isotropic gaussian pixel with a standard deviation of 5 pixels and normalized to have a maximum value of 1.

CA1 activity was then modeled as a rectified sum of random combinations of these inputs. For each CA1 cell, weights for a random 15 inputs were computed by assigning a random weight value on the range [0, 1] for each input and then normalizing all input weights to sum to 1. CA1 cell rate maps were then generated as the weighted sum of these inputs, thresholded at 75% of their maximum value, and smoothed with an isotropic gaussian pixel with a standard deviation of 5 pixels. The resulting modeled CA1 cell rate maps tended to have 1–2 fields in each environment and appeared qualitatively similar to recorded rate maps.

**Histological validation of expression and recording targets**. After experiments, animals were perfused to verify GRIN lens placement. Mice were deeply anesthetized and intracardially perfused with 4% paraformaldehyde in PBS. Brains were dissected and post-fixed with the same fixative. Coronal sections (50 μm) of the entire hippocampus were cut using a vibratome and sections were mounted directly on glass slides. Sections were split and half of all sections were stained for DAPI and mounted with Fluoromount-G (Southern Biotechnology) to localize GRIN lens placement and to evaluate the viral expression. Due to the large imageable surface but restricted miniscope field of view (~0.5 mm × ~0.8 mm), we were unable to determine more specific localization of populations within the hippocampus for mice recorded with 1.8 mm lenses.

**Statistics**. All statistical tests are noted where the corresponding results are reported throughout the main text and supplement. All tests were uncorrected two-tailed tests unless otherwise noted. Z-values for nonparametric Wilcoxon tests were not estimated or reported for comparisons with fewer than 15 data points. Box plots portray the minimum and maximum (whiskers), upper and lower quartiles (boxes), and median (cinch).

**Reporting summary**. Further information on research design is available in the Nature Research Reporting Summary linked to this article.

## Data availability
The data generated in this study have been deposited in the DRYAD database and are accessible at https://doi.org/10.5061/dryad.2z34tmpp9 or via request to the corresponding authors. Source data are provided with this paper.

## Code availability
All custom code written for reported analyses are publicly available at https://github.com/akeinath/MorphExperiments or via request to the corresponding authors.

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

## Acknowledgements

We thank D. Aharoni for extensive guidance in using the UCLA miniscope. We thank J Quinn Lee for helpful feedback on prior versions of this manuscript. During this work, A.T.K. was supported by a McGill University Healthy Brains for Healthy Lives CFREF postdoctoral fellowship and a Natural Sciences and Engineering Research Council (NSERC) Banting postdoctoral fellowship. C.-A.M. was supported by a Fonds de Recherche du Québec—Santé (FRQS) postdoctoral fellowship. Funding was provided by the Canadian Institutes for Health Research (Project grants #367017 and #377074), the Natural Sciences and Engineering Research Council of Canada (Discovery grant #74105), the Canada Research Chairs Program, and the Brain Canada Foundation (Future Leaders in Canadian Brain Science) to M.P.B.

## Author contributions

A.T.K. contributed to experimental design, surgeries, recordings, analysis of data, modeling, as well as drafting and revising of the manuscript. C.-A.M. contributed to surgeries and histology. M.P.B. contributed to experimental design, analysis of data, as well as drafting and revising of the manuscript.

## Competing interests

The authors declare no competing interests.
