## [Peer review file · Nature Communications]

Reviewers' comments:

Reviewer #1 (Remarks to the Author):

This manuscript investigated long-term dynamics of the CA1 spatial code in an extended geometric morph paradigm. The authors demonstrated that on the population level the variance in spatial code correlations can be decomposed into two distinct components: a time component and a spatial context/geometry component. Similarly, variance of spatial code on a single-cell level can also be decomposed into three distinct components: time component, attractor component and geometry component. In addition, they found that variance of spatial code for each cell tends to load heavily on one of the three components, instead of evenly distributed across the three components, suggesting single-cell level selectivity of these components. Lastly, they found that the amount of variance explained by these three components correlate with within session stability of the spatial code. The loadings on these components also interact with other property of the cells such as firing rate and place field preferences.

How CA1 codes for a stable spatial representation while dynamically changing its neural activity patterns across time is a timely topic and of wide interest to the field of neuroscience. The experiments are well-thought and carried out in a very systematic way. The analyses are focused yet complete. However, I had to make a lot of guesses to how the experiments and analyses were conducted. Assuming I made the correct guesses, I think this work could be potentially impactful. However, without a full description of the experimental design and analytical approaches, it is very difficult to judge the overall interpretation, conclusion and impact of the work. I think the descriptions should include enough detail to the extent that a reader can potentially replicate the behavior paradigm and analyses. Otherwise it is really hard to interpret the results. Specifically, I couldn't find information on the following questions:

1. Is "environment A and B" the same as "room A and B"? If so, I don't understand why in fig. 1B the first row (recorded) is labeled as room A. If not, then all other labeling of Room A and B are very confusing.
2. What is the rationale of including unrecorded revisits to the familiar contexts? Is it to further remind the animals of the separate geometric contexts? Would the findings in the paper change if the unrecorded sessions are omitted? What is the time interval between the recorded sessions and each of the corresponding unrecorded revisit sessions?
3. What is the length of each morph sequence? Is it fixed? Do they always have two familiar sessions as overlap? What does "order randomized" mean? Is it truly randomized (non-monotonic in shape space) or only one of the two possible orders (blue to red or red to blue) are randomly chosen each time? Does the order of the two overlap familiar sessions change?

Lack of information really hinders the reader from correctly interpreting the results, especially the modeled RSM in fig. 3A. I think this can be addressed with some dedicated text either in the main result section or in the method section.

Minor comments:

1. Consider making the “morph sequence” labeling more pronounced in fig. 1B (it currently shows up as a light gray text). Also consider making it more clear in the figure that the paradigm continues on for 5 morph sequences. Currently it is easy for the reader to miss that there are 5 “morph sequences” carried out in the pattern shown in fig. 1B. On a related note, consider labeling the “sequences” in fig 1D as “morph sequences 1,2...”, to be both consistent and less ambiguous.
2. In fig. 1D, it is not clear exactly which recording session is used as the “familiar session” and correlated with each session in the 5 morph sequences. Is it the starting/ending “anchoring” session for each sequence? Or is it always the same “pretrained” session? This can greatly influence the interpretation of fig 1D. since time plays an important role for the interpretation.
3. There are multiple instances of strings like “AKCA135”. I assume they are animal name/label, but that’s not always clear from the captions. Consider either labeling them as “Animal 1, 2...” or explicitly state it in the captions.
4. The background data points in fig. 1D are cluttered and do not add information to the plot. If it is intended to show the density/distribution of rate map correlation across cells, consider summarizing them using violin or box plots.
5. I like fig. 2 and I think it is a really nice extension to the description of “representational drift” (Rubin, Ziv et al., 2015). The addition of the geometry dimension fills in a missing piece of the literature and fits with predictions made from the prior literature.
6. In order for reader to interpret fig. 2B, I think it’s critical to understand that it is the RSM matrix “space” that’s being reduced here as 2 dimensions. This information is only clearly stated in the method section, and I don’t think it’s stressed enough in the main text. Currently it is easy to miss what the word “representation” is referring to in the sentence “We then reduced this representation to two dimensions...” And the reader might be misled to think that it is the activity/firing space that’s being reduced.
7. I don’t think there is enough description of the MDS algorithm in the methods. There should at least be some information about how the inputs are transformed and how the embeddings are computed. This information is critical to the interpretation of fig. 2B since depending on the algorithm, orthogonality in the embedding space may or may not be a significant finding.
8. It is expected that the sigmoid change of rate map correlation should be reflected as distinct clustering of red-shaded environments and blue-shaded environments in the MDS reduced plot (fig. 2B). However, by eye that seems to only be true for a few animals. Perhaps it would be an interesting

analysis to quantify the clustering and maybe correlate it with parameters of the fitted sigmoid curve, to establish some correspondence and intuitions between analysis.

9. Since it's critical to understand the construction of expected RSMs to interpret fig. 3, I think there should be more description of these matrices in the main text. Some intuition about why they look like those shown in fig. 3A would be really beneficial.

10. Consider changing the label of "time cells, attractor cells..." in fig. 3D, either to "time, attractors..." as consistent with other plots in fig. 3, or to "time-loading cells, attractor-loading cells" as consistent with the main text. The current labeling could be misleading to some readers.

11. I assume the RSMs shown in fig. 3C are aggregated across all cells in the corresponding loading group. If so, this should be stated more explicitly (like in fig. 2A), since missing this detail can make fig. 3C puzzling to interpret.

12. I don't understand the following interpretation of fig. 3H-K: "Together, these results indicate that short-term spatial reliability and precision is predictive of selectively coding some representational content, but is not predictive of the specific content itself." Does "selectively coding some representational content" refer to significant amount of variances explained by the full model, and "specific content" refer to the GLM loading of each model RSM component? If so, I'm not sure the analysis in this paper is sufficient for such a general interpretation.

Reviewer #2 (Remarks to the Author):

The study by Keinath, Moser, and Brandon used chronic calcium imaging in the hippocampus of mice exploring different (morphed) environments. The goal of the study was to determine whether spatial encoding in CA1 contains distinct representational components (termed: time, attractor, geometry) that differentially change with time. The authors considered two scenarios: (i) The homogenous population-wide representational drift, in which all cells similarly change their encoding over days, irrespective of what they encode (the content of the representation). (ii) That the hippocampus stores time-varying representational component(s) alongside other stable components. This second scenario is said to be more consistent with the so-called "persistent reinstatement hypothesis". To dissociate between these possibilities the authors used a version of a geometric morph paradigm in which 5 mice were imaged through morph sequences of over a total 32 days. The familiarization to the two extreme prototype environments during days 1-7 ensured that the mice will form orthogonal representations of these environments. Indeed, similarly to previous reports, it seems that a portion of the cells abruptly transition between the representations at around the midpoint of the morph sequence (fig. 1). The data presented in figure 2 is also consistent with several previous reports that show that hippocampal representations are dynamic while unique to a given environment. Then, GLM analysis was used to construct for each cell a representational similarity matrix composed of time, attractor and geometry components. My main concern with this work starts here in what seem to me as a circularity in the

analysis (“double dipping”). The authors claim that while time cells change with time, attractor cells and geometry cells don’t. However, this is the only possible outcome for such an analysis in which the cells were initially defined by their tendency to change with time/attractor/geometry. Thus, the authors’ main claim that “CA1 spatial code is defined by distinct representational components with different long-term dynamics, including stable components representing spatial geometry and prior experience” is unsupported by the current analysis. The statements that “Time-loading cells were tracked across more sessions than attractor- or geometry-loading cells (Fig. 4D), consistent with a lack of context-selectivity in these cells” and that “The proportion of each group within the population differentially predicted morph sequence properties: the proportions of geometry- and attractor-loading cells were positively related to the decorrelation between familiar environment representations, while the proportion of time-loading cells was negatively related to this decorrelation (Fig. S2)” are therefore likely an inevitable result of the circularity of the analysis. Overall, while I deem the question of whether there are representational components with different timescales important and timely, the approach taken here to address this question is problematic.

Other issues:

- Given that this study is based on only 5 mice, it is unclear to me why the authors excluded the data from one of the mice in the key analysis. I think that this is unwarranted, especially since this mouse (CAMCA130) clearly shows different cells that actually lie on the different axes (time/attractor/geometry).
- How clean is the time-varying component? In 3 of the 5 mice the number of cells considerably increased with time (mouse AKCA 127 finished the experiment with almost twice as many cells it had in the beginning). Is this a result of virus over-expression? Could this change in cell counts contribute to the time-varying component?
- Likewise, the finding of a trend toward increased decorrelation of maps across sequences (Fig. S2) seem to indicate that learning still occurs in this experiments. This is not surprising in environment with different geometries (e.g. Lever et al 2002), but this could also contribute to the changes in spatial tuning properties over time.
- Details about the GLM analysis are missing from the Methods. For instance, for the time component, what is the time constant of the decay?
- Using the term “time cells” is problematic because it is already associated with a different meaning (e.g. MacDonald et al, 2011).
- Could the relationship between within-session SHC and the full model r^2 simply result from differences in firing rates?

Reviewer #3 (Remarks to the Author):

In this paper, Keinath and colleagues recorded from hundreds of CA1 neurons at the same time while a mouse was navigating in multiple geometrically distinct enclosures. The recordings were carried out using a permanently implanted miniscope which allowed in vivo one-photon imaging in a freely moving mouse. The enclosures included six distinct shapes representing a gradual morphing from a square to a Z-shape environment (although the order of exposure to various shapes was randomised and did not adhere to a gradual transformation). Mice were trained first in two extreme enclosures located in different rooms (Room A and B as stated in Fig1, however in the methods section I did not find the descriptions of experimental rooms. I am also not sure which room was used when recordings/testing began). Then the testing began during which 20 min session was recorded in one of the six enclosures (always starting from 2 'extreme'/baseline shapes) selected at random, with the total number of recorded sessions equal to 32 (recorded over 32 days). The authors showed the CA1 place cells were representing a) previous experiences/attractor states, b) local geometry/boundaries, c) were changing/drifted in time or were unclassified (i.e. did not fit in any of the above criteria). The authors computed median rate map similarity between each rate map of any cell across all sessions and reduced it to two dimensions and showed the existence of these three distinct populations and their mixtures (Fig. 3). As a result, the authors argued that the changes in CA1 place cell activity cannot be explained by a simple homogenous representational drift. Instead, it either represented a selective drift in a specific subpopulation of CA1 place cells or the influence of time-varying inputs.

In summary, I think it is an exciting work and significantly adds to our current understanding of how CA1 can represent different aspects of an animal's environment and time-related information with possible relevance to episodic memory. I found the discussion was not very strong. For example, the authors suggested that '...drift would thus pose a strong challenge to the persistent reinstatement'. However, it is not the case as long as there is a substantial overlap in active place cells between different encounters and that their spatial properties are constant (see point (4) below for details).

I also thought that labelling the place cells as representing prior experiences may be misleading ('prior experience' could mean anything and the geometry of the environment is not necessarily the first thing that comes to mind). A short explanation should be included stating what exactly the authors meant, i.e. the identity of a specific enclosure and that different enclosures had different shapes or similar. 'a representation of the spatial geometry (geometry)' could also be misleading and may be mixed with the first group (i.e. 'prior experience' related group). I would suggest calling it more specifically, e.g. geometric boundaries/configuration/local geometry or similar. I thought that calling the 'representation which varies with time (time)' is the most misleading of all. Especially since it is put on par with other groups which have been much more established; i.e. a number of previous studies showed that local boundaries are represented by CA1 place cell activity, global remapping is also well documented. However, there is no demonstration yet which would have definitively proven that change in place cell activity in time actually encodes time per se; and neither does the current paper.

I also have a few further queries.

1) I was concerned about the lack of stability in their place cells (at least the ones shown in their examples, e.g. Fig. 1 and Supplementary Figure 3 which I assume shows their best cells from each class): Fig. 1C cells seemed incredibly unstable on pre and post (i.e. two baseline enclosures) – only cells 2&7 were stable, similar concerns apply to the Supplementary Figure 3. That attractor cells in general were

quite unstable in red sequences and very stable at blue sequences (firing at the corner). Could the authors suggest why? This is not in line with the previous studies, e.g. Dombeck (2010) using two photon imaging or Ziv et al (2013) using 1p imaging. It would be important to show the ROIs (i.e. 1p imaging) of simultaneously recorded hippocampal cells across days to make sure that their ROI identification is reliable; Supplementary Figure 1 is not adequate (I could barely see any individual cell).

2) I could not understand the rationale of taking the median similarity value across all pair-wise correlations for each individual cell and also doing this for the population activity analysis? Could the authors explain?

3) Fig S1 – the number of cells goes up in three out of 5 animals while the number of tracked cells go down – why is it so and is this consistent with previous reports?

4) 'However, longitudinal experiments in mice have challenged this view^{11,12}, reporting that the vast majority of CA1 place cells change their spatial tuning properties more rapidly than expected (on the order of days), while only a subset of these cells (15-25%) provide a stable spatial representation on this timescale¹²': this sentence is misleading as it may imply that 15-25% of CA1 place cells are stable and the vast majority are not, whereas in reality 15-25% of cells remain active from one experience to the next encountered 5 days later while other previously silent place cells become active. Some cells overlap between different experiences and when they are active the vast majority of them have stable unchanged place field locations. In other words, this rapid change in most of the cases refers to changes in place cell firing rates rather than global remapping or location drift. As a result, what the authors called 'drift' would not pose any decoding problem as long as there is enough overlap between active population (in relation to the size of the environment) on any given day.

5) Traces in Fig 1A are hard to see – so it was hard to assess how well their deconvolution algorithm performed. The authors should consider overlaying the 'spikes' from the deconvolution algorithm with the Ca traces and making that whole sub-figure bigger.

6) All the experiments were carried out in the light part of the light/dark cycle. Was it not a problem for the behaviour? Can they show the behaviour plots (i.e. that the sampling was adequate etc).

7) Why every mouse had a different dilution of GCaMP6 virus?

8) The room illumination was different in the training sessions (well-lit room) vs recording sessions (only the light from the monitor was on). Why? Could this affect the results?

We are grateful to all the reviewers for their appraisals and in-depth critiques of our initial submission. On the basis of this feedback we have made substantial changes to every section of the manuscript, including: A complete revision of the introduction and discussion sections, replacement of the previous single-cell analysis with an alternative approach which we believe addresses the critiques of the initial submission, a more thorough characterization of within-morph-sequence single cell results, the addition of computational modeling results to situate the results of our dimensionality reduction analysis among possible alternatives, and the inclusion of a decoding analysis which demonstrates the generalizability of the contextual code across weeks-long timescales despite ongoing representational changes. In most of these cases these changes were in direct response to particular critiques; however, given the extent of the changes made, some specific questions are no longer directly relevant to the revised manuscript. Below we provide more detail about how our revisions address each of the points raised.

Reviewer #1 (Remarks to the Author):

This manuscript investigated long-term dynamics of the CA1 spatial code in an extended geometric morph paradigm. The authors demonstrated that on the population level the variance in spatial code correlations can be decomposed into two distinct components: a time component and a spatial context/geometry component. Similarly, variance of spatial code on a single-cell level can also be decomposed into three distinct components: time component, attractor component and geometry component. In addition, they found that variance of spatial code for each cell tends to load heavily on one of the three components, instead of evenly distributed across the three components, suggesting single-cell level selectivity of these components. Lastly, they found that the amount of variance explained by these three components correlate with within session stability of the spatial code. The loadings on these components also interact with other property of the cells such as firing rate and place field preferences.

How CA1 codes for a stable spatial representation while dynamically changing its neural activity patterns across time is a timely topic and of wide interest to the field of neuroscience. The experiments are well-thought and carried out in a very systematic way. The analyses are focused yet complete. However, I had to make a lot of guesses to how the experiments and analyses were conducted. Assuming I made the correct guesses, I think this work could be potentially impactful. However, without a full description of the experimental design and analytical approaches, it is very difficult to judge the overall interpretation, conclusion and impact of the work. I think the descriptions should include enough detail to the extent that a reader can potentially replicate the behavior paradigm and analyses. Otherwise it is really hard to interpret the results. Specifically, I couldn't find information on the following questions:

We thank the reviewer for their valued feedback.

1. Is "environment A and B" the same as "room A and B"? If so, I don't understand why in fig. 1B the first row (recorded) is labeled as room A. If not, then all other labeling of Room A and B are very confusing.

We recognize that these details create more confusion than necessary and have removed the room A/B designation from Figure 1. These details are now instead listed in the Methods.

2. What is the rationale of including unrecorded revisits to the familiar contexts? Is it to further remind the animals of the separate geometric contexts? Would the findings in the paper change if the unrecorded sessions are omitted? What is the time interval between the recorded sessions and each of the corresponding unrecorded revisit sessions?

The reviewer is correct that the motivation for including additional unrecorded experience was to further remind the mouse of the separate geometric contexts. Given the duration of the experiment our initial concern was that the initial familiarization might not be sufficient to drive sustained discrimination of the familiar environments and/or morph dynamics. However, newly available data from (Plitt & Giocomo, 2021) suggest that the initial familiarization in morph experiments such as these does indeed have a sustained influence, and as such we expect that the primary results would obtain even in the absence of continued additional experience, perhaps with the exception of greater map decorrelation across sequences. The details concerning how the unrecorded experience was implemented are now listed in the methods.

3. What is the length of each morph sequence? Is it fixed? Do they always have two familiar sessions as overlap? What does "order randomized" mean? Is it truly randomized (non-monotonic in shape space) or only one of the two possible orders (blue to red or red to blue) are randomly chosen each time? Does the order of the two overlap familiar sessions change?

We have made the following changes in the revision to clarify these paradigmatic questions: For each within-sequence analysis we state whether the analysis relied on a 6-day (include only the first set of familiar environments) or 8-day (include a second set of bookending familiar environments) morph sequence. 8-day morph sequence analyses did always overlap with two familiar sessions between neighboring sequences, 6-day analyses did not overlap with any data. The order of the 4 morph environments was truly random on each sequence, and we have altered the schematic in Fig. 1b to convey this. The order of the two overlap familiar sessions did not change within each mouse, but was randomized across mice. The methods have been updated to include these clarifications as well.

Lack of information really hinders the reader from correctly interpreting the results, especially the modeled RSM in fig. 3A. I think this can be addressed with some dedicated text either in the main result section or in the method section.

We have removed this analysis from the revision and have replaced it with a conceptually-related analysis to address concerns raised by other reviewers. This new analysis directly addresses the relationship between contextual coding and long-term stability within individual cells while controlling for covariates in firing rate in a more interpretable fashion (see the new figure 5).

Minor comments:

1. Consider making the “morph sequence” labeling more pronounced in fig. 1B (it currently shows up as a light gray text). Also consider making it more clear in the figure that the paradigm continues on for 5 morph sequences. Currently it is easy for the reader to miss that there are 5 “morph sequences” carried out in the pattern shown in fig. 1B. On a related note, consider labeling the “sequences” in fig 1D as “morph sequences 1,2...”, to be both consistent and less ambiguous.

We have changed the figure to include this information.

2. In fig. 1D, it is not clear exactly which recording session is used as the “familiar session” and correlated with each session in the 5 morph sequences. Is it the starting/ending “anchoring” session for each sequence? Or is it always the same “pretrained” session? This can greatly influence the interpretation of fig 1D. since time plays an important role for the interpretation.

We have updated the figure now corresponding to this analysis (Fig. 2a) to explicitly include the days contributing to each morph-sequence analysis to eliminate this ambiguity.

3. There are multiple instances of strings like “AKCA135”. I assume they are animal name/label, but that’s not always clear from the captions. Consider either labeling them as “Animal 1, 2...” or explicitly state it in the captions.

We have now updated the figure captions to highlight that these strings are the mouse names.

4. The background data points in fig. 1D are cluttered and do not add information to the plot. If it is intended to show the density/distribution of rate map correlation across cells, consider summarizing them using violin or box plots.

We agree that the background datapoints were cluttering and did not add much. We have replaced these plots with more interpretable mean +/- SEM options.

5. I like fig. 2 and I think it is a really nice extension to the description of “representational drift” (Rubin, Ziv et al., 2015). The addition of the geometry dimension fills in a missing piece of the literature and fits with predictions made from the prior literature.

We appreciate the reviewer highlighting the contribution of this finding.

6. In order for reader to interpret fig. 2B, I think it’s critical to understand that it is the RSM matrix “space” that’s being reduced here as 2 dimensions. This information is only clearly stated in the method section, and I don’t think it’s stressed enough in the main text. Currently it is easy to miss what the word “representation” is referring to in the sentence “We then reduced this representation to two dimensions...” And the reader might be misled to think that it is the activity/firing space that’s being reduced.

We have now updated this section of the text to highlight the data contributing to this analysis, with the specific language: *To this end, we first computed the mean rate map similarity across all tracked cells for each pairwise comparison of sessions (Fig. 2a). These matrices exhibited structural features which*

generalized across mice: mean rate map correlations vary from high values for pairs of sessions close in time toward zero for sessions pairs separated further in time, with additional modulation by contextual similarity. To make this structure explicit...

7. I don't think there is enough description of the MDS algorithm in the methods. There should at least be some information about how the inputs are transformed and how the embeddings are computed. This information is critical to the interpretation of fig. 2B since depending on the algorithm, orthogonality in the embedding space may or may not be a significant finding.

We now include additional information about the specific implementation of the MDS in the methods section. Moreover, our revision includes computational modeling results to help situate the findings we report among alternative structures which differ in their drift dynamics. These modeling results help make clear the diverse structures possible with this analysis, further highlighting the significance of the structure we observe in our actual data.

8. It is expected that the sigmoid change of rate map correlation should be reflected as distinct clustering of red-shaded environments and blue-shaded environments in the MDS reduced plot (fig. 2B). However, by eye that seems to only be true for a few animals. Perhaps it would be an interesting analysis to quantify the clustering and maybe correlate it with parameters of the fitted sigmoid curve, to establish some correspondence and intuitions between analysis.

In our revision we now include the sigmoid change of rate map correlations (a.k.a. transition plots) for all mice in the main text Figure 2a, as well as the MDS for all mice in Fig. 3b, allowing the reader to more clearly assess the relationship between the two, as highlighted by the reviewer here. The reader now has complete information to note a general correspondence between the clustering of context in the MDS and the extent of sigmoidal dynamics (both the extent of map decorrelations and the abruptness of the transition dynamics) as suggested by the reviewer.

9. Since it's critical to understand the construction of expected RSMs to interpret fig. 3, I think there should be more description of these matrices in the main text. Some intuition about why they look like those shown in fig. 3A would be really beneficial.

As noted above, we have now updated this section of the text to provide some intuition about these matrices, with the specific language: *To this end, we first computed the mean rate map similarity across all tracked cells for each pairwise comparison of sessions (Fig. 2a). These matrices exhibited structural features which generalized across mice: mean rate map correlations vary from high values for pairs of sessions close in time toward zero for sessions pairs separated further in time, with additional modulation by contextual similarity. To make this structure explicit...*

10. Consider changing the label of "time cells, attractor cells..." in fig. 3D, either to "time, attractors..." as consistent with other plots in fig. 3, or to "time-loading cells, attractor-loading cells" as consistent with the main text. The current labeling could be misleading to some readers.

The revised analysis compares continuous measures of drift and contextual factors, and no longer results in a categorical labeling of cells which led to this issue.

11. I assume the RSMs shown in fig. 3C are aggregated across all cells in the corresponding loading group. If so, this should be stated more explicitly (like in fig. 2A), since missing this detail can make fig. 3C puzzling to interpret.

This was indeed the case, though this presentation has been removed from the revision in place of the new single-cell analysis.

12. I don't understand the following interpretation of fig. 3H-K: "Together, these results indicate that short-term spatial reliability and precision is predictive of selectively coding some representational content, but is not predictive of the specific content itself." Does "selectively coding some representational content" refer to significant amount of variances explained by the full model, and "specific content" refer to the GLM loading of each model RSM component? If so, I'm not sure the analysis in this paper is sufficient for such a general interpretation.

This was indeed the interpretation that we had in mind; however, we agree that such an interpretation is better suited as a discussion possibility rather than a direct conclusion in the results. Our revised single-cell analysis focuses more specifically on the relationship between interpretable contextual coding and long-term stability, and this possibility is only referenced in the discussion section.

Reviewer #2 (Remarks to the Author):

The study by Keinath, Moser, and Brandon used chronic calcium imaging in the hippocampus of mice exploring different (morphed) environments. The goal of the study was to determine whether spatial encoding in CA1 contains distinct representational components (termed: time, attractor, geometry) that differentially change with time. The authors considered two scenarios: (i) The homogenous population-wide representational drift, in which all cells similarly change their encoding over days, irrespective of what they encode (the content of the representation). (ii) That the hippocampus stores time-varying representational component(s) alongside other stable components. This second scenario is said to be more consistent with the so-called “persistent reinstatement hypothesis”. To dissociate between these possibilities the authors used a version of a geometric morph paradigm in which 5 mice were imaged through morph sequences of over a total 32 days. The familiarization to the two extreme prototype environments during days 1-7 ensured that the mice will form orthogonal representations of these environments. Indeed, similarly to previous reports, it seems that a portion of the cells abruptly transition between the representations at around the midpoint of the morph sequence (fig. 1). The data presented in figure 2 is also consistent with several previous reports that show that hippocampal representations are dynamic while unique to a given environment. Then, GLM analysis was used to construct for each cell a representational similarity matrix composed of time, attractor and geometry components. My main concern with this work starts here in what seem to me as a circularity in the analysis (“double dipping”). The authors claim that while time cells change with time, attractor cells and geometry cells don’t. However, this is the only possible outcome for such an analysis in which the cells were initially defined by their tendency to change with time/attractor/geometry. Thus, the authors’ main claim that “CA1 spatial code is defined by distinct representational components with different long-term dynamics, including stable components representing spatial geometry and prior experience” is unsupported by the current analysis. The statements that “Time-loading cells were tracked across more sessions than attractor- or geometry-loading cells (Fig. 4D), consistent with a lack of context-selectivity in these cells” and that “The proportion of each group within the population differentially predicted morph sequence properties: the proportions of geometry- and attractor-loading cells were positively related to the decorrelation between familiar environment representations, while the proportion of time-loading cells was negatively related to this decorrelation (Fig. S2)” are therefore likely an inevitable result of the circularity of the analysis. Overall, while I deem the question of whether there are representational components with different timescales important and timely, the approach taken here to address this question is problematic.

We thank the reviewer for their valued feedback. In light of the feedback concerning the single-cell analyses in the initial submission, we have now eschewed that analysis in favor of a more direct assay of the relationship between contextual coding and long-term stability. Like in the initial submission, this analysis indicated that cells whose representational similarity resembled a contextual code also tended to exhibit greater long-term stability. Crucially, in this revised analysis we cross-validated this analysis such that separate data contributed to the characterization of the contextual code and to the characterization of long-term dynamics to ensure that these results were not driven by ‘double-dipping’-style artifacts. Moreover, this revised analysis further allowed us to control for firing rate covariates, indicating that the relationship between interpretable contextual coding and long-term stability goes beyond what would be expected on the basis of firing rates alone. Finally, we have revised the results and discussion section to ensure that interpretive claims are presented in the discussion alone, and balanced with alternative interpretations acknowledged.

Other issues:

- Given that this study is based on only 5 mice, it is unclear to me why the authors excluded the data from one of the mice in the key analysis. I think that this is unwarranted, especially since this mouse (CAMCA130) clearly shows different cells that actually lie on the different axes (time/attractor/geometry).

In the initial submission, CAMCA130 was only excluded from one aggregate figure (Fig. 3c), but did contribute to all other analyses. In the revision, all animals contributed to all analyses – no data were excluded.

- How clean is the time-varying component? In 3 of the 5 mice the number of cells considerably increased with time (mouse AKCA 127 finished the experiment with almost twice as many cells it had in the beginning). Is this a result of virus over-expression? Could this change in cell counts contribute to the time-varying component?

While the number of cells does increase in 3 of the 5 mice over time (which may be driven by changes to viral expression, or by recruitment of a larger population of cells with continued experience), the results from these mice do match those of the other 2 mice with stable cell counts over the course of the experiment (Fig. 3b), indicating that the time-varying component is not driven by the cell counts alone.

- Likewise, the finding of a trend toward increased decorrelation of maps across sequences (Fig. S2) seem to indicate that learning still occurs in this experiments. This is not surprising in environment with different geometries (e.g. Lever et al 2002), but this could also contribute to the changes in spatial tuning properties over time.

We agree with the reviewer's interpretation of this finding. In the revision we also include an analysis which demonstrates that accurate contextual information can be read out from the population even across long timescales despite these ongoing representational changes, indicating that differences due to changes in cell counts or learning across large lags (e.g. 4 sequences) do not irretrievably distort the contextual representation.

- Details about the GLM analysis are missing from the Methods. For instance, for the time component, what is the time constant of the decay?

We have replaced this analysis with an alternative approach.

- Using the term "time cells" is problematic because it is already associated with a different meaning (e.g. MacDonald et al, 2011).

We agree with the reviewer and have removed this nomenclature from the revised manuscript.

- Could the relationship between within-session SHC and the full model r^2 simply result from differences in firing rates?

As noted above, we now control for firing rate differences in our revised analysis.

Reviewer #3 (Remarks to the Author):

In this paper, Keinath and colleagues recorded from hundreds of CA1 neurons at the same time while a mouse was navigating in multiple geometrically distinct enclosures. The recordings were carried out using a permanently implanted miniscope which allowed in vivo one-photon imaging in a freely moving mouse. The enclosures included six distinct shapes representing a gradual morphing from a square to a Z-shape environment (although the order of exposure to various shapes was randomised and did not adhere to a gradual transformation). Mice were trained first in two extreme enclosures located in different rooms (Room A and B as stated in Fig1, however in the methods section I did not find the descriptions of experimental rooms. I am also not sure which room was used when recordings/testing began). Then the testing began during which 20 min session was recorded in one of the six enclosures (always starting from 2 'extreme'/baseline shapes) selected at random, with the total number of recorded sessions equal to 32 (recorded over 32 days). The authors showed the CA1 place cells were representing a) previous experiences/attractor states, b) local geometry/boundaries, c) were changing/drifted in time or were unclassified (i.e. did not fit in any of the above criteria). The authors computed median rate map similarity between each rate map of any cell across all sessions and reduced it to two dimensions and showed the existence of these three distinct populations and their mixtures (Fig. 3). As a result, the authors argued that the changes in CA1 place cell activity cannot be explained by a simple homogenous representational drift. Instead, it either represented a selective drift in a specific subpopulation of CA1 place cells or the influence of time-varying inputs.

In summary, I think it is an exciting work and significantly adds to our current understanding of how CA1 can represent different aspects of an animal's environment and time-related information with possible relevance to episodic memory. I found the discussion was not very strong. For example, the authors suggested that '...drift would thus pose a strong challenge to the persistent reinstatement'. However, it is not the case as long as there is a substantial overlap in active place cells between different encounters and that their spatial properties are constant (see point (4) below for details).

We thank the reviewer for their valued feedback. In our revision we have rewritten the introduction and discussion to better establish the existing perspectives on the relationship between representational content and drift, as well as to clarify the relationship between the current results and this debate in a more balanced manner.

I also thought that labelling the place cells as representing prior experiences may be misleading ('prior experience' could mean anything and the geometry of the environment is not necessarily the first thing that comes to mind). A short explanation should be included stating what exactly the authors meant, i.e. the identity of a specific enclosure and that different enclosures had different shapes or similar. 'a representation of the spatial geometry (geometry)' could also be misleading and may be mixed with the first group (i.e. 'prior experience' related group). I would suggest

calling it more specifically, e.g. geometric boundaries/configuration/local geometry or similar. I thought that calling the 'representation which varies with time (time)' is the most misleading of all. Especially since it is put on par with other groups which have been much more established; i.e. a number of previous studies showed that local boundaries are represented by CA1 place cell activity, global remapping is also well documented. However, there is no demonstration yet which would have definitively proven that change in place cell activity in time actually encodes time per se; and neither does the current paper.

We agree that this nomenclature could lend itself to confusions and misinterpretations, and have eliminated this categorization from the revision. We have replaced this with a new analysis to more specifically address the relationship between contextual representation and long-term stability at the level of individual cells in a way that avoids potential confounds such as firing rate differences.

I also have a few further queries.

1) I was concerned about the lack of stability in their place cells (at least the ones shown in their examples, e.g. Fig. 1 and Supplementary Figure 3 which I assume shows their best cells from each class): Fig. 1C cells seemed incredibly unstable on pre and post (i.e. two baseline enclosures) – only cells 2&7 were stable, similar concerns apply to the Supplementary Figure 3. That attractor cells in general were quite unstable in red sequences and very stable at blue sequences (firing at the corner). Could the authors suggest why? This is not in line with the previous studies, e.g. Dombeck (2010) using two photon imaging or Ziv et al (2013) using 1p imaging. It would be important to show the ROIs (i.e. 1p imaging) of simultaneously recorded hippocampal cells across days to make sure that their ROI identification is reliable; Supplementary Figure 1 is not adequate (I could barely see any individual cell).

We have replaced these examples with clearer, more stable example cells, and have included their ROIs for reference (Fig. 1c).

2) I could not understand the rationale of taking the median similarity value across all pair-wise correlations for each individual cell and also doing this for the population activity analysis? Could the authors explain?

We have now replaced the single-cell analysis in our resubmission with a different analysis that does not include this averaging. Moreover, we now show that similar results are obtained when population vector correlations are used as the measure of similarity between pairwise session comparisons (Fig. S3, and below):

Thus the results generalize across difference measures of population-level similarity.

3) Fig S1 – the number of cells goes up in three out of 5 animals while the number of tracked cells go down – why is it so and is this consistent with previous reports?

While the number of cells does increase in 3 of the 5 mice over time (which may be driven by changes to viral expression given our surgical timeline, or by recruitment of a larger population of cells with continued experience), the results from these mice do match those of the other 2 mice with stable cell counts over the course of the experiment (Fig. 3b), indicating that the results we report are not a result of mere increases in the number of cells. Decreases in the number of tracked cells with larger passages of time is a common observation, i.e. (Ziv et al., 2013; Fig 3c below), with our tracking results within the range typically observed.

4) 'However, longitudinal experiments in mice have challenged this view^{11,12}, reporting that the vast majority of CA1 place cells change their spatial tuning properties more rapidly than expected (on the order of days), while only a subset of these cells (15-25%) provide a stable spatial representation on this timescale¹²': this sentence is misleading as it may imply that 15-25% of CA1 place cells are stable and the vast majority are not, whereas in reality 15-25% of cells remain active from one experience to the next encountered 5 days later while other previously silent place cells become active. Some cells overlap between different experiences and when they are active the vast majority of them have stable unchanged place field locations. In other words, this rapid change in most of the cases refers to changes in place cell firing rates rather than global remapping or location drift. As a result, what the authors called 'drift' would not pose any decoding problem as long as there is enough overlap between active population (in relation to the size of the environment) on any given day.

We have changed the language with which these previous findings are discussed to be more specific about these findings, and have noted that primarily rate-driven changes have been observed in some cases as the reviewer notes here (Ziv et al., 2013), while other reports include progressive changes in field location in addition to rate changes (e.g. Lee et al., 2020; Fig. 2b):

5) Traces in Fig 1A are hard to see – so it was hard to assess how well their deconvolution algorithm performed. The authors should consider overlaying the 'spikes' from the deconvolution algorithm with the Ca traces and making that whole sub-figure bigger.

We have altered this figure to increase the visibility of the calcium traces. In the revision we have used a different method to extract the rising phase of calcium transients as our measure of spiking activity. We found that this method was less affected by slow fluctuations in baseline fluorescence which could contaminate the deconvolved signal, though all results were ultimately similar regardless of which spiking assay we used.

6) All the experiments were carried out in the light part of the light/dark cycle. Was it not a problem for the behaviour? Can they show the behaviour plots (i.e. that the sampling was adequate etc).

This was not a problem for behavior, as mice maintained more than adequate coverage of the environments across the entirety of the experiment. Below is a plot of proportion of the environment visited for at least 0.5 sec for each session for all mice:

Coverage was generally thorough and remind similarly high for the duration of recording sessions. Thorough coverage was likely encouraged by the dim ambient lighting, as noted below. For additional reference, below is the raw trajectory for AKCA135:

and the raw trajectory for AKCA130:

7) Why every mouse had a different dilution of GCaMP6 virus?

The mice recorded in these experiments came from a surgical cohort in which we were testing various dilutions of a new batch of virus, which yielded viable mice at multiple dilutions. We considered mice viable when expression levels were high (many cells visible and active) but no pathology was observed that is typical with over-expression in CA1 (constantly 'active' cells, slow moving calcium waves across the FOV).

8) The room illumination was different in the training sessions (well-lit room) vs recording sessions (only the light from the monitor was on). Why? Could this affect the results?

The illumination was reduced during recording to encourage the mice to sample the space more completely, which was the case as noted above. While dimmer, ambient lighting was sufficient to view the surrounding environment and cues, and was not equivalent to true 'darkness' in the typical experimental sense where any light is eliminated. While we cannot rule out an influence of the difference in lighting on the results, measures of place field quality such as split-half rate map correlations and spatial information content were similar to well-lit open field recordings in a similarly-sized environment, indicating that the dim ambient lighting did not qualitatively impair fundamental spatial coding properties in these experiments.

REVIEWER COMMENTS

Reviewer #1 (Remarks to the Author):

The authors made substantial improvements on the manuscript and have addressed my comments very well. In particular, I like the addition of simulation results presented in fig. 3 c-e, which gives important intuition of the results and their relationship with biological hypothesis. I think this is very valuable for these types of complex analyses. I also appreciated the decoding work presented in fig. 4, which addresses a very important question about representational drift. Overall, I think the present manuscript will be impactful in the field and contribute substantially to our understanding of representational drift.

My major suggestions is about which analysis to include across the current and original version of the manuscript. To me, the single-cell sigmoid fit analysis presented in fig. 1 d-f doesn't add much to the story. The result basically indicates that firing activity of cells can be interpreted as coding for the two extreme/familiar context, and such coding are different across cells in terms of some parameters. I think this adds little to the theme of how spatial map evolve across time in the presence of representational drift. I recommend combining fig. 1 and fig. 2 (which is basically fig. 1 in the original manuscript) which presents the basic single-cell characterizations in a much more compact way. Also, I think the GLM analysis in fig. 3 of the original manuscript is a clever way to investigate the coding of spatial context vs time on a single cell level, and with slight modification (focusing only on context/attractor vs time) the GLM analysis can serve as a very good complement to the current single cell analysis in fig. 5. I recommend including that GLM analysis in this manuscript, even if it's in the supplemental figures.

There were points in the manuscript where I was confused on the details so it would be helpful to clarify these points below.

1. I am confused by the equations referencing sigmoid functions. I tried plotting the function in the presented form and I get a hyperbola shape instead of S-shape. Furthermore the parameter p_2 doesn't seem to control the sharpness/abruptness of the curve. It seems both the constant 1 and p_2 are misplaced.

2. I am confused by how the 2d sigmoid are modeled and fitted. Since the RSM is by definition symmetric, I don't see why the authors choose to convert the sigmoid function to 2d in a symmetric way, instead of simply aggregating the data to 1d. Similarly, the comparison to 5-parameter polynomial doesn't make sense to me either. Again since the RSM is symmetric, I don't see how p_1 can be different from p_3 (or p_2 from p_4) in the polynomial model. Overall, it seems to me that the author

are just comparing the fit of a sigmoid with the fit of a second order polynomial, for which I don't understand the rationale.

3. In fig. 1e, the fact that the most dominant/expected cluster (1) are in the center, and the most "extreme" clusters (9, 10, 11, 12) basically all have the transition point at the end of the shape space makes me question the point of the UMAP analysis and how it's relevant to the theme of the manuscript. It would be nice to see results from linear dimension reduction approaches, so that we know how much the distances/projections are driven by single parameters (for example, p_1), and whether that's surprising at all.

4. I cannot find how exactly the decoder is formulated for fig. 4. I can guess from the main text that a single sequence is used as a training set, and the prediction is basically argmax of the correlation to the training set. However, information about the way the predictions are aggregated as "proportion correct", and whether and how cross-validation are carried out, are all missing from the text. It would be beneficial to include a detailed formulation of the decoder analysis in methods section.

5. It would be interesting to see the same decoder analysis (fig. 4) carried out with all the cells included, where a cell missing from a session would have an activity of 0. Although this might not add much to the main point of the paper, I think this is an important analysis to have since turnover of cells is an important aspect of representational drift.

Reviewer #3 (Remarks to the Author):

The authors made some substantial changes to the main text, figures and some of the major analysis methods, although the results and conclusions of the paper remained largely the same. Namely, that CA1 place cells can be split into three major groups: 1) the ones which represent a global drift in time from their initial presentation; 2) the ones which are more stable either encoded context (more square-like vs. more polygon like) or 3) geometry/boundaries of the enclosure. The authors argued that the findings suggest that the context information (note not so much the geometry-related information) can be reliably (~80% accurate) decoded over a 32-day period without any need for an adoptive 'read-out' system, while the global drift signal may reflect some information about a particular experience or even time. In my previous review, I argued that the authors cannot prove that the global drift indeed encodes time or anything to do with the episode and they changed the discussions to moderate their claim. I believe the current discussions are indeed more balanced and justified.

In general, the major rationale for the substantial changes the authors made was to address the potential circularity of their former approach pointed out by one of the reviewers as well as to moderate and balance the conclusions that they may draw based on their results. As I stated in my first reply, I believe that overall the study is timely and exciting. However, as in the previous version, I have some concerns the authors should address prior to the publication:

1) They should include the information on animal's behaviour in the Supplementary Materials so that Readers can make their own judgments whether the coverage is adequate. From what the authors showed in the responses (i.e. the percentage of bins visited for at least 0.5 s during the trial) the sampling may not be sufficient: a mouse AKCA127 and AKCA 170 do not seem to be adequate (the authors did not show their raw coverage). They should also provide the correlations coefficients of rate maps recorded during the first and the second parts of the trial. They simply mentioned that the correlations are high.

2) What does 'Extracted spatial footprints' refers to?

3) The following statement was confusing: '(Fig. 2a). These matrices exhibited structural features which generalized across mice: mean rate map correlations vary from high values for pairs of sessions close in time toward zero for sessions pairs separated further in time, with additional modulation by contextual similarity.' I am not sure how Fig 2a represents temporal information unless the authors mean different sequences. If so, I am not sure I see high correlation values close in time vs further away in time. Which values do they have in mind?

4) I do not understand how Fig. 2b represents a contextual vs. drift component.

5) As in the previous version of the paper it is still not clear to me what the drift is referenced to (i.e. what is taken as a baseline)? It should be clearly stated in the main text. Is it the baseline recorded before each sequence?

6) The authors should also describe a bit better what is MDS and how it is calculated to provide better intuition. The same applies to UMAP. These concepts are much less standard and not necessarily familiar to a common Reader.

7) Why do model MDS often go in the opposite direction from the measured MDS? AKCA 170 seemed like a contextual drift to me (only going in the opposite direction).

8) Finally, the notion that there are other categories of cells have almost entirely disappeared from this version of the article. I think this should be mentioned in the conclusions.

In addition, the editor asked me to comment on the response of the authors to three of Reviewer 2's comments from the previous round of review. Below are my comments on three outstanding problems:

1) The authors' main claim that "CA1 spatial code is defined by distinct representational components with different long-term dynamics, including stable components representing spatial geometry and prior experience" is unsupported by the current analysis because of circular arguments.

The authors addressed the circularity of the argument by changing their initial analysis methods to more single-cell based approach where they calculated RMS of individual cells and approximated them with 5-parameter sigmoidal description. They then performed

Uniform Manifold Approximation and Projection (UMAP) to reduce the 5-dimension characterization to two dimensions and performed k-medoids clustering with $k = 12$ (number 12 was chosen arbitrary likely just representing a relatively high number presumably so that different functional cells are not clustered together). Fig. 1f shows typical RMS of cells from each cluster: e.g. cluster 1 corresponds to place cells showing a classical attractor dynamics responses as previously described by Wills et al. 2005, cluster 12 represent more or less stable cells and other clusters most likely represent different cell groups which were active in some environments but silent in others (it is hard to tell for sure because the authors do not show representative rate maps from each RMS of the k-group). So far, I think the findings are consistent with what was previously reported and do not suffer from circularity or double-dipping. However, when it comes to definitions of time-drifting cells it is still circular and not clear how so called time-drifting cells relate to contextual cells – I still could not find any indication that they are separate vs. the same cell populations. In other words, it would amount to a trivial conclusion to suggest that something that is not stable 'represents' time vs something that is stable does not. The question is how much cells representing the context or geometry also drift in time – it still was not clear in the new manuscript because pretty much the entire analysis after Fig.1a-d is done on the population of cells (i.e. all of the cells taken together for each animal separately; on that note, I would also like to add that the authors never specified whether all of the included cells were place cells and how the place cell criterion was defined).

3) How clean is the time-varying component? In 3 of the 5 mice the number of cells considerably increased with time (mouse AKCA 127 finished the experiment with almost twice as many cells it had in the beginning). Is this a result of virus over-expression? Could this change in cell counts contribute to the time-varying component?

This point is probably even more problematic than the point above. Firstly, I am not sure whether their decorrelation measure is even the most appropriate measure to indicate the 'change in time'. It is a very specific 'change in time' related to the transition point in the morphing experiment (at least this is my guess since the authors' description of what it is was very brief and quite ambiguous, the schematic would certainly be appreciated). In any case, even using this measure the authors point to Fig. 2c as evidence for continuous representational time drift: "In all cases, the familiar and morph environments remained

partially correlated with one another, though we observed a continues decorrelation across morph sequences such that the final morph sequence showed significantly more decorrelation than the first morph sequence in all mice (Fig. 2c). This gradual decorrelation might be a result of experience with the morph sequence itself or the ongoing daily unrecorded top-up experience in each of the familiar environments. Together, these results demonstrate that our adapted geometric morph paradigm replicates many phenomena observed within morph sequence in acute versions of this paradigm, and extends these findings to demonstrate that these within sequence dynamics evolve with continued experience.” However, I don’t think that Fig. 2c supports this statement. Namely, the decorrelation is not monotonically increasing in time but rather goes up and then down again (in the case of 4 out of 5 animals). The authors did not discuss what this could mean and how this would affect their interpretation.

Moreover, I also did not think that the data supported the authors claim that the temporal dynamics in two mice with stable cell numbers were consistent with the other 3 animals. In my opinion, Fig 3 a-b indicate quite the opposite. Fig. 3a suggests that in these two animals (two graphs from the right) RMS appeared to be more stable in time compared to the other three animals. Also, Fig. 3b MDS analysis suggests that these animals showed contextual group drift vs. global drift seen in the other 3 animals which may have indeed been caused by input from newly appearing cells. It is just really hard to interpret their results since everything is shown on the population level and it is hard to know what this really means in terms of place cell coding.

On the minor point: the authors are referring to Fig 2a 2b etc on the bottom page 6 and page 7 when they describe Fig. 3a, 3b etc.

4) Likewise, the finding of a trend toward increased decorrelation of maps across sequences (Fig. S2) seem to indicate that learning still occurs in this experiments. This is not surprising in environment with different geometries (e.g. Lever et al 2002), but this could also contribute to the changes in spatial tuning properties over time.

I also agree with the reviewer and am not quite sure how to rule this out. Neither did the authors and tried to address this indirectly by saying that the contextual and environment (i.e. geometry) information can be read in time relatively accurately hence a reasonable degree of stability at least for context decoding was present. I don’t think that this answer rules out the alternative interpretation suggested by Referee 2 and I would just simply suggest including this explanation as one of the possibilities accounting for the drift.

Reviewer #1 (Remarks to the Author):

The authors made substantial improvements on the manuscript and have addressed my comments very well. In particular, I like the addition of simulation results presented in fig. 3 c-e, which gives important intuition of the results and their relationship with biological hypothesis. I think this is very valuable for these types of complex analyses. I also appreciated the decoding work presented in fig. 4, which addresses a very important question about representational drift. Overall, I think the present manuscript will be impactful in the field and contribute substantially to our understanding of representational drift.

My major suggestions is about which analysis to include across the current and original version of the manuscript. To me, the single-cell sigmoid fit analysis presented in fig. 1 d-f doesn't add much to the story. The result basically indicates that firing activity of cells can be interpreted as coding for the two extreme/familiar context, and such coding are different across cells in terms of some parameters. I think this adds little to the theme of how spatial map evolve across time in the presence of representational drift. I recommend combining fig. 1 and fig. 2 (which is basically fig. 1 in the original manuscript) which presents the basic single-cell characterizations in a much more compact way. Also, I think the GLM analysis in fig. 3 of the original manuscript is a clever way to investigate the coding of spatial context vs time on a single cell level, and with slight modification (focusing only on context/attractor vs time) the GLM analysis can serve as a very good complement to the current single cell analysis in fig. 5. I recommend including that GLM analysis in this manuscript, even if it's in the supplemental figures.

We thank the reviewer for their constructive criticism and feedback. We have now taken the reviewers advice in our revision, combining the within-sequence analyses from the prior Fig 1 + 2 into a single figure, removing the UMAP analysis from the main text and instead substituting a similar analysis in the supplement, and including a version of the GLM analysis focusing on context vs drift to complement the other single cell-level analysis.

There were points in the manuscript where I was confused on the details so it would be helpful to clarify these points below.

1. I am confused by the equations referencing sigmoid functions. I tried plotting the function in the presented form and I get a hyperbola shape instead of S-shape. Furthermore the parameter p_2 doesn't seem to control the sharpness/abruptness of the curve. It seems both the constant 1 and p_2 are misplaced.

We thank the reviewer for catching this mistranslation of the code into the print equation. We have now corrected this and tested it to ensure that the reported function behaves as described.

2. I am confused by how the 2d sigmoid are modeled and fitted. Since the RSM is by definition symmetric, I don't see why the authors choose to convert the sigmoid function to 2d in a symmetric way, instead of simply aggregating the data to 1d. Similarly, the comparison to 5-parameter polynomial doesn't make sense to me either. Again since the RSM is symmetric, I don't see how p_1 can be different from p_3 (or p_2 from p_4) in the polynomial model. Overall, it seems to me that the author are just comparing the fit of a sigmoid with the fit of a second order polynomial, for which I don't understand the rationale.

We apologize for the lack of clarity in the fitting of contextual RSMs. We agree with the reviewer that the contextual RSMs are by definition symmetric across the diagonal. Because of this, we fit only the upper triangle of each RSM, and mirrored this fit for presentation purposes. We now include this detail in the methods. Furthermore, we agree with the reviewer that the rational for the polynomial fit comparison is tangential to the claims made for this analysis, and have removed this comparison from the text.

3. In fig. 1e, the fact that the most dominant/expected cluster (1) are in the center, and the most "extreme" clusters (9, 10, 11, 12) basically all have the transition point at the end of the shape space makes me question the point of the UMAP analysis and how it's relevant to the theme of the manuscript. It would be

nice to see results from linear dimension reduction approaches, so that we know how much the distances/projections are driven by single parameters (for example, p_1), and whether that's surprising at all.

We agree with the reviewer that the addition of the UMAP analysis was tangential to the main thrust of the manuscript. As such, we have removed it from the main text, and included a similar analysis in the supplement (Fig. S9) instead. Here, we used Isomap (less complex/initialization-dependent than UMAP) to produce our 2D embeddings from our 5 parameter model fits. To highlight the relationship between individual model parameters and the resulting structure, we now color-code the embedding to indicate how each model parameter relates to this structure. We also include example RSMs and note where they lie in this embedding.

4. I cannot find how exactly the decoder is formulated for fig. 4. I can guess from the main text that a single sequence is used as a training set, and the prediction is basically argmax of the correlation to the training set. However, information about the way the predictions are aggregated as “proportion correct”, and whether and how cross-validation are carried out, are all missing from the text. It would be beneficial to include a detailed formulation of the decoder analysis in methods section.

We now include these details in the main text when the decoder is introduced, as follows: “Specifically, we took the sessions from a given six-day morph sequence and a target session outside of the predictor set, and predicted the contextual group and exact environment identity from the best matching predictor set session. Then we repeated this process for all target sessions and using each six-day morph sequence as the predictor set once, and aggregating the results according to the lag between the target session and the nearest predictor set session.”

5. It would be interesting to see the same decoder analysis (fig. 4) carried out with all the cells included, where a cell missing from a session would have an activity of 0. Although this might not add much to the main point of the paper, I think this is an important analysis to have since turnover of cells is an important aspect of representational drift.

We now report the outcome for both treatments of missing/inactive cells in the main text (Fig. 3b,c). The decoding results are similar regardless of whether we include only tracked cells or if we substitute zeroed-maps for missing cells, though the raw population vector correlations are slightly lower with the zero-substitution result:

Reviewer #3 (Remarks to the Author):

The authors made some substantial changes to the main text, figures and some of the major analysis methods, although the results and conclusions of the paper remained largely the same. Namely, that CA1 place cells can be split into three major groups: 1) the ones which represent a global drift in time from their initial presentation; 2) the ones which are more stable either encoded context (more square-like vs. more polygon like) or 3) geometry/boundaries of the enclosure. The authors argued that the findings suggest that the context information (note not so much the geometry-related information) can be reliably (~80% accurate) decoded over a 32-day period without any need for an adoptive 'read-out' system, while the global drift signal may reflect some information about a particular experience or even time. In my previous review, I argued that the authors cannot prove that the global drift indeed encodes time or anything to do with the episode and they changed the discussions to moderate their claim. I believe the current discussions are indeed more balanced and justified. In general, the major rationale for the substantial changes the authors made was to address the potential circularity of their former approach pointed out by one of the reviewers as well as to moderate and balance the conclusions that they may draw based on their results. As I stated in my first reply, I believe that overall the study is timely and exciting. However, as in the previous version, I have some concerns the authors should address prior to the publication:

We thank the reviewer for their fair appraisal and constructive feedback.

1) They should include the information on animal's behaviour in the Supplementary Materials so that Readers can make their own judgments whether the coverage is adequate. From what the authors showed in the responses (i.e. the percentage of bins visited for at least 0.5 s during the trial) the sampling may not be sufficient: a mouse AKCA127 and AKCA 170 do not seem to be adequate (the authors did not show their raw coverage).

We now include examples of the behavioral sampling as well as a quantification of all sessions for all animals in the supplement, with a more stringent minimum sampling per pixel of 1 s for this quantification. For reference, at our spatial binning, perfectly even sampling would result in 4.69 s of sampling per pixel. Furthermore, in order to ensure that sampling differences between environments and across time cannot account for any of the effects we report, we now match the sampling distributions by subsampling our data prior to computing our measures (described in the methods, as well as in [Keinath et al., 2018 eLife; Keinath et al., 2020, Nat Comm]).

They should also provide the correlations coefficients of rate maps recorded during the first and the second parts of the trial. They simply mentioned that the correlations are high.

We now include the distributions of split-half correlation values and spatial information content values, as well as their significance relative to shuffled controls, for all animals and sessions in the supplementary material.

2) What does 'Extracted spatial footprints' refers to?

This refers to the spatial component of the matrix factorization which corresponds to the location and shape of each cell extracted in the field of view, which we now clarify in the main text.

3) The following statement was confusing: '(Fig. 2a). These matrices exhibited structural features which generalized across mice: mean rate map correlations vary from high values for pairs of sessions close in time toward zero for sessions pairs separated further in time, with additional modulation by contextual similarity.' I am not sure how Fig 2a represents temporal information unless the authors mean different sequences. If so, I am not sure I see high correlation values close in time vs further away in time. Which values do they have in mind?

This was a mistaken figure reference which we have now corrected in this revision.

4) I do not understand how Fig. 2b represents a contextual vs. drift component.

This was a mistaken figure reference which we have now corrected in this revision.

5) As in the previous version of the paper it is still not clear to me what the drift is referenced to (i.e. what is taken as a baseline)? It should be clearly stated in the main text. Is it the baseline recorded before each sequence?

We now state in the introduction that neural drift refers to the long-timescale changes to the CA1 spatial code observed even across repeated visits to the same environment. In the results we now clarify how drift is operationalized where relevant.

6) The authors should also describe a bit better what is MDS and how it is calculated to provide better intuition. The same applies to UMAP. These concepts are much less standard and not necessarily familiar to a common Reader.

We have now described in more detail in both the main text and methods the motivation, computations, and interpretations one can draw from this analysis. For example, in the main text: "MDS is a set of unsupervised techniques for transforming a potentially high-dimensional pairwise distance matrix into a set of points in a low-dimensional space while preserving the relative distances between points as well as possible, as defined by some cost function. In the case of nonmetric MDS, the cost function aims to preserve the rank order relationships between points but tolerates a nonparametric monotonic scaling of distances, which is useful when measured distances are not expected to scale linearly as is the case with correlation distances. A strength of MDS is its applicability to cases where the relative similarity between conditions is measured, not the raw position of each condition in a common feature space, as is the case here. Thus, in the current application nMDS will map the mean pairwise rate map similarity matrix to a two-dimensional embedding where each session is a point and the relative position of sessions indicates their similarity, while the raw scale and overall orientation of the arrangement are arbitrary."

At the suggestion of reviewer 1, we have now replaced the UMAP analysis from the main text with a similar but simpler and less initialization-dependent Isomap analysis in the supplement, as it was tangential to the main thrust of the manuscript (Fig. S9). To provide the reader with an intuition for how this embedding relates to the underlying model parameters (rather than arbitrarily clustering this embedding) we color-coded the embedding according to each model parameter. We also include example RSMs and note where they lie in this embedding.

7) Why do model MDS often go in the opposite direction from the measured MDS? AKCA 170 seemed like a contextual drift to me (only going in the opposite direction).

As we now clarify in our introduction of the technique, MDS aims to preserve the relative distances between points in the low-dimensional embedding; however, the orientation and raw scale of the output are arbitrary (the consistency observed across some of the animals is a product of nMDS being initialized by a metric MDS PCA-like process. If the measurements between sessions were Euclidean distances in a common coordinate frame rather than correlation values, metric MDS would be equivalent to PCA).

8) Finally, the notion that there are other categories of cells have almost entirely disappeared from this version of the article. I think this should be mentioned in the conclusions.

On the advice of Reviewer 1, we now include a version of the GLM analysis motivating these original claims.

In addition, the editor asked me to comment on the response of the authors to three of Reviewer 2's comments from the previous round of review. Below are my comments on three outstanding problems:

1) The authors' main claim that "CA1 spatial code is defined by distinct representational components with different long-term dynamics, including stable components representing spatial geometry and prior experience" is unsupported by the current analysis because of circular arguments.

The authors addressed the circularity of the argument by changing their initial analysis methods to more single-cell based approach where they calculated RMS of individual cells and approximated them with 5-parameter sigmoidal description. They then performed Uniform Manifold Approximation and Projection (UMAP) to reduce the 5-dimension characterization to two dimensions and performed k-medoids clustering with $k = 12$ (number 12 was chosen arbitrary likely just representing a relatively high number presumably so that different functional cells are not clustered together). Fig. 1f shows typical RMS of cells from each cluster: e.g. cluster 1 corresponds to place cells showing a classical attractor dynamics responses as previously described by Wills et al. 2005, cluster 12 represent more or less stable cells and other clusters most likely represent different cell groups which were active in some environments but silent in others (it is hard to tell for sure because the authors do not show representative rate maps from each RMS of the k-group). So far, I think the findings are consistent with what was previously reported and do not suffer from circularity or double-dipping. However, when it comes to definitions of time-drifting cells it is still circular and not clear how so called time-drifting cells relate to contextual cells – I still could not find any indication that they are separate vs. the same cell populations. In other words, it would

amount to a trivial conclusion to suggest that something that is not stable ‘represents’ time vs something that is stable does not.

As suggested by the feedback here, we now include both a version of the GLM analysis as described in the original manuscript and the contextual RSM-based analysis from our prior revision. These two analyses complement one another in provide evidence for a relationship between contextual coding and long-term stability. In the GLM analysis (Fig. 4), we now include a shuffled control to evaluate the significance of the heterogeneity among individual cells in context-loading versus drift-loading variance. Here we find many cells with significant context-attributable variance but not drift-attributable variance in their similarity across sessions, and vice versa, speaking directly to this question about the relationship between the two. Moreover, we agree with the reviewer that the suggestion that drift-attributable variance is evidence for a representation of ‘time’ is not a defensible claim; while consistent with such a view, these data are also consistent with many other interpretations. We now use ‘drift’ language instead of ‘time’ language to help avoid this confusion, and highlight the variety of interpretations with which these data are consistent in the discussion.

The question is how much cells representing the context or geometry also drift in time – it still was not clear in the new manuscript because pretty much the entire analysis after Fig.1a-d is done on the population of cells

In addition to the insights from the GLM analysis speaking to this question, in the revision we also speak to this question in the contextual RSM vs long-term stability analysis now in Fig. 5. Here we group cells according to their evidence for interpretable contextual coding and quantify their long-term stability (controlling for sampling differences between comparisons and firing rate differences between groups). We now group our data into sextiles instead of quartiles for higher resolution (though the outcome is qualitatively the same for all groupings we applied on the range of 3 to 8 groups). Here we see that cells with the most evidence for contextual coding are

significantly more stable than cells with less evidence of contextual coding across all lags, and that these highly-contextual cells maintain relatively high stability values even across a lag of four sequences.

Moreover, to give the reader a better intuition about how the contextual RSMs we characterize in this analysis relate to the underlying rate maps, and a calibration for interpreting the raw stability values in terms of these rate maps, we now include examples of contextual RSMs in the supplement (Fig. S9). These examples are purposefully diverse in their patterns and noisiness.

(i.e. all of the cells taken together for each animal separately; on that note, I would also like to add that the authors never specified whether all of the included cells were place cells and how the place cell criterion was defined).

We now note that: “Because of variability in whether the same cell met traditional within-session place cell criteria across sessions, and to avoid any biases that subselecting cells based on functional tuning properties might introduce, all cells were eligible for inclusion regardless of their spatial tuning during a given session. We now also include a supplementary figure speaking to this result (Fig. S5).”

3) How clean is the time-varying component? In 3 of the 5 mice the number of cells considerably increased with time (mouse AKCA 127 finished the experiment with almost twice as many cells it had in the beginning). Is this a result of virus over-expression? Could this change in cell counts contribute to the time-varying component?

This point is probably even more problematic than the point above. Firstly, I am not sure whether their decorrelation measure is even the most appropriate measure to indicate the 'change in time'. It is a very specific 'change in time' related to the transition point in the morphing experiment (at least this is my guess since the authors' description of what it is was very brief and quite ambiguous, the schematic would certainly be appreciated). In any case, even using this measure the authors point to Fig. 2c as evidence for continuous representational time drift: "In all cases, the familiar and morph environments remained partially correlated with one another, though we observed a continuous decorrelation across morph sequences such that the final morph sequence showed significantly more decorrelation than the first morph sequence in all mice (Fig. 2c). This gradual decorrelation might be a result of experience with the morph sequence itself or the ongoing daily unrecorded top-up experience in each of the familiar environments. Together, these results demonstrate that our adapted geometric morph paradigm replicates many phenomena observed within morph sequence in acute versions of this paradigm, and extends these findings to demonstrate that these within sequence dynamics evolve with continued experience." However, I don't think that Fig. 2c supports this statement. Namely, the decorrelation is not monotonically increasing in time but rather goes up and then down again (in the case of 4 out of 5 animals). The authors did not discuss what this could mean and how this would affect their interpretation.

We appreciate the attention brought to these concerns. We agree with the reviewer that these data could have been described more precisely, though we are wary of interpreting the non-monotonic nature of this result as it is unclear whether we should expect within-mouse monotonicity across sequences given potential measurement variability. We have now reworded this description to: "In all cases, the familiar and morph environments remained partially correlated with one another, though final morph sequences were significantly more decorrelated than the first morph sequence across mice (Fig. 1f). This increased decorrelation might be a result of ongoing learning from experience with the morph sequence itself and/or the daily unrecorded top-up experience in each of the familiar environments. Altogether, these results demonstrate that our adapted geometric morph paradigm replicates many phenomena observed within-sequence in acute versions of this paradigm and suggests that these dynamics may continue to evolve with experience."

Moreover, I also did not think that the data supported the authors claim that the temporal dynamics in two mice with stable cell numbers were consistent with the other 3 animals. In my opinion, Fig 3 a-b indicate quite the opposite. Fig. 3a suggests that in these two animals (two graphs from the right) RMS appeared to be more stable in time compared to the other three animals. Also, Fig. 3b MDS analysis suggests that these animals showed contextual group drift vs. global drift seen in the other 3 animals which may have indeed been caused by input from newly appearing cells.

Although there is some variability in the extent of stability across mice, in all cases we do see a decorrelation of the maps even when measured in the same environment across time. As one measure of this, in the revised version of our decoding analysis, we now report the correlation value between the best matching predictor set day and the target day as a function of lag (i.e. time between the predictor set and target day) for two different treatments of missing/inactive cells (See below). In all cases and for all mice we see a consistent decrease in the best matching correlation value across time. Furthermore, we now clarify the interpretation of the nMDS plots to note that factors which clearly differ across animals, such as raw orientation and scale, are arbitrary, as only the relative arrangement among sessions and not its absolute embedding can be interpreted. Finally, to ensure that our results are not driven by the increasing cell counts in a subset of the animals across time, (in addition to matching spatial sampling distributions) we now also match the number of cells included in each pairwise comparison within animal by randomly subsampling tracked cells for this analysis. Our results remain qualitatively unchanged even

when we do so. We also include the nMDS results for two measures of similarity (mean rate map correlation and population vector correlation), and when using nMDS to compute a three-dimensional embedding, all of which produce qualitatively similar results.

It is just really hard to interpret their results since everything is shown on the population level and it is hard to know what this really means in terms of place cell coding.

We now include both a version of our initial GLM analysis as well as the analysis of interpretable contextual coding versus long-term stability at the single-cell level, with example cells to help provide an intuition for how these analyses relate to the underlying rate maps. More generally, we focus on population-level measures and aggregated single-cell analyses because such measures are a strength a calcium imaging, where the size of the aggregated datasets can offset the more variable single-cell-signal that we observe with imaging as opposed to electrophysiological techniques.

On the minor point: the authors are referring to Fig 2a 2b etc on the bottom page 6 and page 7 when they describe Fig. 3a, 3b etc.

We thank the reviewer for highlighting this mistaken reference, we have now corrected this.

4) Likewise, the finding of a trend toward increased decorrelation of maps across sequences (Fig. S2) seem to indicate that learning still occurs in this experiments. This is not surprising in environment with different geometries (e.g. Lever et al 2002), but this could also contribute to the changes in spatial tuning properties over time.

I also agree with the reviewer and am not quite sure how to rule this out. Neither did the authors and tried to address this indirectly by saying that the contextual and environment (i.e. geometry) information can be read in time relatively accurately hence a reasonable degree of stability at least for context decoding was present. I don't think that this answer rules out the alternative interpretation suggested by Referee 2 and I would just simply suggest including this explanation as one of the possibilities accounting for the drift.

As the reviewer suggests we now include statements in the results and discussion stating that ongoing learning might be one source of the long-term changes in neural activity which we observe.

REVIEWERS' COMMENTS

Reviewer #1 (Remarks to the Author):

The authors made significant improvements on the manuscript and have addressed all my comments very well. I think the presented work will be impactful for the field and contribute substantially to our understanding of representational drift.

I only have 2 minor suggestions:

1. The caption of fig.1 mentioned inferred firing rates in 1a. However, I can only find the filtered calcium traces but not firing rates in 1a.
2. In fig. 4c, it might be beneficial to label the horizontal axis as “context” and vertical axis as “time”, so that it’s easier for the reader to know what the two example cells convey.

The authors made significant improvements on the manuscript and have addressed all my comments very well. I think the presented work will be impactful for the field and contribute substantially to our understanding of representational drift.

I only have 2 minor suggestions:

1. The caption of fig.1 mentioned inferred firing rates in 1a. However, I can only find the filtered calcium traces but not firing rates in 1a.

We thank the reviewer for catching this, and have corrected this figure caption.

2. In fig. 4c, it might be beneficial to label the horizontal axis as “context” and vertical axis as “time”, so that it’s easier for the reader to know what the two example cells convey.

We thank the reviewer for this suggestion, and have altered the figure caption to note that these examples are presented chronologically, in order of recording.